# Permutation-Consistent Variational Encoding for Incomplete Multi-View Multi-Label Classification

**Chengliang Liu[1]**    **Bo Li[1]**    **Bob Zhang[1]***    **Xiaoling Luo[2]**    **Yabo Liu[3]**    **Jie Wen[4]***

[1] University of Macau
[2] Shenzhen University    [3] Ocean University of China
[4] Harbin Institute of Technology, Shenzhen

liucl1996@163.com    bo_li_mail@163.com    bobzhang@um.edu.mo
xiaolingluoo@outlook.com    yaboliu.ug@gmail.com    jiewen_pr@126.com

## Abstract

Incomplete multi-view multi-label learning is fundamentally an information integration problem under simultaneous view and label incompleteness. We introduce Permutation-Consistent Variational Encoding framework (PCVE) with an information bottleneck strategy, which learns variational representations capable of aggregating shared semantics across views while remaining robust to incompleteness. PCVE formulates a principled objective that maximizes a variational evidence lower bound to retain task-relevant information, and introduces a permutation-consistent regularization to encourage distributional consistency among representations that encode the same target semantics from different views. This regularization acts as an information alignment mechanism that suppresses view-private redundancy and mitigates over-alignment, thereby improving both sufficiency and consistency of the learned representations. To address incompleteness, PCVE further incorporates a masked multi-label learning objective that leverages available supervision. Extensive experiments across diverse benchmarks and missing ratios demonstrate consistent gains over state-of-the-art methods in multi-label classification, while enabling reliable inference of missing views without explicit imputation. Analyses corroborate that the proposed information-theoretic formulation improves cross-view semantic cohesion and preserves discriminative capacity, underscoring the effectiveness and generality of PCVE for incomplete multi-view multi-label learning.

## 1 Introduction

Multi-view multi-label learning has become a central paradigm for modeling complex entities that are naturally described by heterogeneous sources (e.g., image–text pairs, multi-sensor signals, or multimodal clinical records) and annotated with multiple, potentially correlated labels. By jointly exploiting complementary and redundant information across views, multi-view learning methods can efficiently enhance semantic coverage, reduce ambiguity, and capture high-level discriminative feature for downstream tasks Fang et al. (2022); Qin et al. (2025a); Fang et al. (2025). However, real-world deployments rarely enjoy complete data: views and labels are frequently missing due to acquisition failures, privacy constraints, or cost. Under such inevitable view and label incompleteness, extracting and aggregating shared information across views becomes markedly challenging due to unreliable cross-view alignment, dominated view-private noise, and sparsely informative supervision, degrading representation sufficiency and predictive performance Qin et al. (2025b); Li et al. (2025a). In this paper, we define this multi-view learning task with double missing issue as incomplete multi-view missing multi-label classification (iM3C).

---

*Corresponding authors

Despite recent advances, multi-view learning still struggles with the core goal of representation sufficiency, i.e., joint embeddings should retain as much task-relevant information as is shared across views while discarding view-private nuisance Qin & Qian (2024). Non-probabilistic deep methods, typically based on contrastive learning or InfoMax objectives, often depend on model architecture and the chosen estimators for good results; they help in practice but do not provide clear guarantees about sufficiency. Probabilistic and information-theoretic methods are able to explicit model two key goals, i.e., retaining shared information and removing nuisance information Li et al. (2025b), However, many existing information theory-based approaches implement only single-variable consistency in the shared latent space, i.e., maximizing pairwise dependence or aligning multi-view data and cross-view representation. Such scalar constraints are coarse, providing neither sufficiency guarantees nor protection against contamination from not fully trained or low-quality views. That may lead to the information aggregation process in joint representation learning being affected by insufficiently learned views (i.e., information redundancy or learning collapse).

In this work, we enforce cross-view alignment early in the encoding stage within an information bottleneck (IB) formulation, encouraging compact yet sufficient shared representations. Concretely, we employ cross-view encoders where each view's observations condition a shared latent distribution cluster, enabling decoupled per-view inference while supporting distribution alignment across views. To strengthen semantic consistency, we introduce a permutation-consistency objective that exchanges distributions of latent variables corresponding to different views, to regularize cross-view matching with scalable complexity. In parallel, we incorporate view-specific reconstruction terms that preserve view-valid information and prevent over-compression of task-relevant content. These components are unified under a principled IB-style objective with a variational derivation, for which we provide a complete variational derivation.

Our contributions are summarized as follows:

- We propose a universal variational encoding framework for incomplete multi-view multi-label classification that accommodates arbitrary patterns of view and label incompleteness, while learning deep semantic consistency from constrained observations.

- We develop a permutation-consistency empowered IB model that exploits permutation invariance of cross-view representations to impose view-wise consistency constraints, which maximizes task-relevant information extraction while retaining view-specific content as much as possible.

- We present extensive empirical evidence demonstrating that the proposed framework consistently outperforms strong baselines and achieves state-of-the-art results in both missing and fully observed regimes.

## 2 PRELIMINARY

### 2.1 INCOMPLETE MULTI-VIEW MISSING MULTI-LABEL CLASSIFICATION

Multi-view learning leverages both redundancy and complementarity across views to improve robustness and downstream performance. Early approaches emphasize consistency and complementarity regularization (e.g., co-training and kernel alignment), and later evolved toward deep representation learning and semi-supervised regimes to cope with missing observations and noise in practice (Andrew et al., 2013; Liu et al., 2022; 2020). For incomplete multi-view settings (missing views), dominant strategies include: matrix/tensor completion with low-rank priors to recover cross-view structure (Wen et al., 2019); cross-view alignment and shared–private factorization to disentangle common semantics from view-specific factors (Liu et al., 2023a; Lin et al., 2024); and deep contrastive or consistency-based methods that maintain discriminative representations despite missing views (Luo et al., 2024; Bian et al., 2024). Under missing multi-label supervision (partially observed labels), existing methods typically combine label-dependency modeling with self-training, extrapolating unobserved labels via graph regularization, conditional dependencies, or deep pseudo-label estimation (Xie et al., 2024; Chen et al., 2019; Li et al., 2024). Overall, strategies for handling either incomplete multi-view learning or missing multi-label classification are diverse, however, research that simultaneously addresses both forms of incompleteness has only recently emerged. Representative work includes: DICNet, which first introduces multi-view contrastive learning into the iM3C

task and achieves significant performance gains (Liu et al., 2023b); AIMNet, which generates imputed views via graph-based neighbor retrieval (Liu et al., 2024a), and NAIM3L, which cleverly employs dual-index information to mitigate the adverse effects of missing views and labels (Li & Chen, 2023).

## 2.2 Information theory-based multi-view learning

Variational autoencoder (VAE) (Pu et al., 2016) provides a probabilistic framework for information theory-based multi-view representation learning, which is commonly used to unify cross-modal generation, missing-data completion, and uncertainty quantification under shared latent variables (Wan et al., 2021). Early multi-view VAEs posit a joint latent space to explain multiple sources, thereby enabling coherent generation and inference across views (Vedantam et al., 2018; Wu & Goodman, 2018). Representative lines of work include jointly inference-based multimodal VAEs that fuse multiple sources with a shared latent space Vedantam et al. (2018); Wu & Goodman (2018); Khattar et al. (2019), posterior aggregation via product of experts (PoE) or mixture of experts (MoE), enabling robust inference when arbitrary modality subsets are observed (Qiu et al., 2025; Chakrabarty & Pal, 2024; Tan et al., 2024), and alignment or mutual-information regularization to strengthen semantic sharing and identifiability (Liu et al., 2024b). For incomplete inputs, a key focus is conditional posteriors and marginal generation to support inference and reconstruction from any subset of views (Liao et al., 2022; Chen et al., 2025).

## 2.3 Problem formulation

Consider an incomplete multi-view multi-label dataset $\mathcal{D} = \{(\mathbf{x}, \mathbf{y})\}$ with $N$ samples. Each instance is described by $m$ views, denoted $\mathbf{x} = \{\mathbf{x}^{(v)}\}_{v=1}^{m}$ with $\mathbf{x}^{(v)} \in \mathbb{R}^{d_v}$. Due to view incompleteness, we set $\mathcal{V}$ with $|\mathcal{V}| \leq m$ to denote the set of observed views for any instance. The label assignment is given by $\mathbf{y} \in \{0,1\}^C$, where $C$ is the number of categories and $\mathbf{y}^c = 1$ indicates the membership in class $c$. Similarly, we define $\mathcal{G}$ with $|\mathcal{G}| \leq C$ as the available label set for any instance. Our goal is to encoding the joint multi-view representation $\mathbf{z}$ given the incomplete data $\mathcal{D}$ with prior missing information $\mathcal{V}$ and $\mathcal{G}$, and classify the instance into corresponding categories. Note that in the context of information theory, we use random variables to describe data and problem definitions.

## 2.4 Sufficiency of multi-view representations

When is a multi-view representation "good enough" for multi-label prediction? Our answer builds on a simple intuition and turns it into a practical, trainable target. Different views of the same sample describe the same subject: they may look different in low-level details, but the task-relevant meaning should be consistent. With this in mind, we adopt a semantic consistency assumption:

**Assumption 2.1** *For the prediction of* $\mathbf{y}$*, there exists semantic information that is shared among multiple views* $\mathbf{x}^{(1)}, \mathbf{x}^{(2)}, \ldots, \mathbf{x}^{(m)}$*. Then, for mutual information between views and target, we have:*

$$I(\mathbf{x}^{(1)}; \mathbf{y}) = I(\mathbf{x}^{(2)}; \mathbf{y}) = \ldots = I(\mathbf{x}^{(m)}; \mathbf{y})$$

This assumption intuitively demonstrates that each available view $\mathbf{x}^{(v)}$ carries essentially the same information about the target $\mathbf{y}$. Unlike a single-view setting, the multi-view setting provides more chances to filter out content irrelevant to the task. Therefore, following previous work (Liu et al., 2024b), we introduce joint multi-view representation $\mathbf{z}$ to associate multiple views in the embedding space and state a direct proposition:

**Proposition 2.2** *If multi-view joint representation* $\mathbf{z}$ *contains all the information shared by all views,* $\mathbf{z}$ *is sufficient for predicting* $\mathbf{y}$*.*

An appropriately constructed cross-view representation $\mathbf{z}$ can, in principle, contain the information required for downstream prediction. However, raw multi-view observations usually contain substantial view-specific variability that is not aligned with the target. Thus naive fusion often carries these irrelevant factors into $\mathbf{z}$, introducing redundancy that obscures the shared task-relevant signal and weakens the predictive efficacy of $\mathbf{z}$ for $\mathbf{y}$ (Federici et al., 2020; Liu et al., 2024b). These consid-

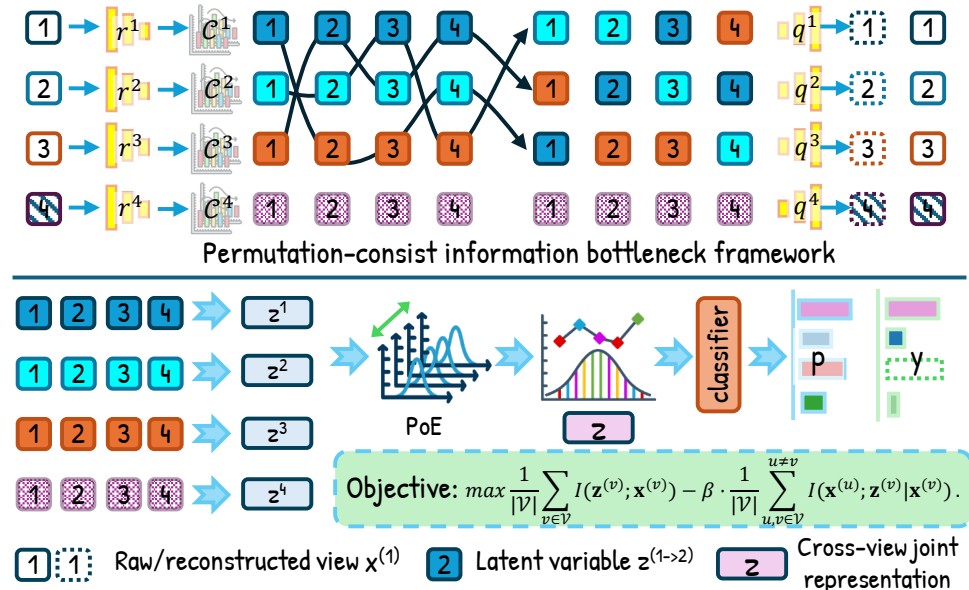

Figure 1: Our main framework of PCVE. The multi-view shared information learning and reconstruction is shown at the top; the cross-view fusion and multi-label classification process is in the bottom.

erations motivate learning strategies that emphasize shared semantics while suppressing non-shared information, thereby maintaining the sufficiency of $\mathbf{z}$ under realistic conditions.

## 3 METHODOLOGY

In this section, we detail the proposed permutation-consistent variational encoding (PCVE) framework. The overall pipeline is illustrated in Fig. 1: the top panel depicts multi-view shared information learning and reconstruction, while the bottom panel shows cross-view fusion and multi-label classification.

### 3.1 CONSISTENCY INFORMATION BOTTLENECK FRAMEWORK

Building on Proposition 2.2, a shared representation $\mathbf{z}$ that captures cross-view commonality is semantically sufficient for prediction. Further constraining the model to remove all non-shared redundancy, ensuring that $\mathbf{z}$ retains only shared information, will be more beneficial for downstream tasks. Concretely, for given views $\mathbf{x}^{(v)}$ and $\mathbf{x}^{(u)}$, consider the mutual information between $\mathbf{x}^{(u)}$ and joint variables $\mathbf{z}$ and $\mathbf{x}^{(v)}$, we have:

$$I(\mathbf{x}^{(u)}; \mathbf{x}^{(v)}, \mathbf{z}) = I(\mathbf{x}^{(u)}; \mathbf{z} \mid \mathbf{x}^{(v)}) + I(\mathbf{x}^{(v)}; \mathbf{x}^{(u)}) = I(\mathbf{x}^{(v)}; \mathbf{x}^{(u)} \mid \mathbf{z}) + I(\mathbf{z}; \mathbf{x}^{(u)})$$

$$\Rightarrow I(\mathbf{x}^{(u)}; \mathbf{z} \mid \mathbf{x}^{(v)}) = \underbrace{I(\mathbf{z}; \mathbf{x}^{(u)}) - I(\mathbf{x}^{(v)}; \mathbf{x}^{(u)})}_{\text{Minimum sufficiency}} + \underbrace{I(\mathbf{x}^{(v)}; \mathbf{x}^{(u)} \mid \mathbf{z})}_{\text{Sufficiency}}. \tag{1}$$

This decomposition subject to two assumptions. First, if $\mathbf{z}$ contains the shared information across views, then the mutual information between any views is equal to 0, i.e., $I(\mathbf{x}^{(v)}; \mathbf{x}^{(u)} \mid \mathbf{z}) = 0$. Second, if $\mathbf{z}$ contains only the shared information, then the mutual information between $\mathbf{z}$ and any view is equivalent to the shared part between views, i.e., $I(\mathbf{z}; \mathbf{x}^{(u)}) - I(\mathbf{x}^{(v)}; \mathbf{x}^{(u)}) = 0$. Under these two conditions, we can get $I(\mathbf{x}^{(u)}; \mathbf{z} \mid \mathbf{x}^{(v)}) = 0$, which formalizes the notion of minimal sufficiency with respect to the cross-view common semantics.

**Corollary 3.1** *For any pair of views $u \neq v$, let $\mathbf{z}$ be a joint multi-view representation that contains and only contains the shared information across views, then the conditional mutual information $I(\mathbf{x}^{(u)}; \mathbf{z} \mid \mathbf{x}^{(v)}) = 0$.*

To attain minimal sufficiency of $\mathbf{z}$ while maximizing shared information, it suffices to minimize the conditional dependence between views given $\mathbf{z}$, i.e., to drive $I(\mathbf{x}^{(u)}; \mathbf{x}^{(v)} \mid \mathbf{z})$ toward zero. Previous work typically represents the multi-view joint coding with a single variable $\mathbf{z}$ and models its distribution from raw multi-view data using PoE or MoE, while imposing a posterior-consistency constraint during fusion. This strategy implicitly assumes a one-to-one mapping between each view and its view-specific latent component, e.g., the component of $\mathbf{z}$ attributed to view $v$ solely originates from $\mathbf{x}^{(v)}$. Although intuitive, such fusion can suffer from view imbalance and training insufficiency: some dominant views may steer the PoE/MoE aggregation and overshadow weaker views. We therefore propose to impose the cross-view consistency constraint earlier, specifically at the stage of view-specific distribution modeling, to suppress non-shared information. Concretely, we introduce explicit view-specific components $\{\mathbf{z}^{(v)}\}_{v \in \mathcal{V}}$ as a decomposition of the joint representation $\mathbf{z}$, and replace the learning objective with a sum of conditional terms $\frac{1}{|\mathcal{V}|} \sum_{v \in \mathcal{V}} I(\mathbf{x}^{(u)}; \mathbf{z}^{(v)}|\mathbf{x}^{(v)})$. Intuitively, minimizing $I(\mathbf{x}^{(u)}; \mathbf{z}^{(v)}|\mathbf{x}^{(v)})$ helps to compress $\mathbf{z}^{(v)}$ to remove redundant view-specific information. However, minimizing this term alone risks information collapse in $\{\mathbf{z}^{(v)}\}$, yielding degenerate non-informative representations. Therefore, we complement that objective with an informativeness regularizer that preserves the valid information within raw data, by maximizing $I(\mathbf{z}^{(v)}; \mathbf{x}^{(v)})$ while simultaneously minimizing the conditional mutual information:

$$\max \ \frac{1}{|\mathcal{V}|} \sum_{v \in \mathcal{V}} I\big(\mathbf{z}^{(v)}; \mathbf{x}^{(v)}\big), \quad \text{s.t.} \ \frac{1}{|\mathcal{V}|} \sum_{u,v \in \mathcal{V}}^{u \neq v} I\big(\mathbf{x}^{(u)}; \mathbf{z}^{(v)}|\mathbf{x}^{(v)}\big) = 0. \tag{2}$$

Introducing a Lagrange multiplier $\beta \geq 0$ yields the unconstrained objective:

$$\max \ \frac{1}{|\mathcal{V}|} \sum_{v \in \mathcal{V}} I\big(\mathbf{z}^{(v)}; \mathbf{x}^{(v)}\big) \ - \ \beta \cdot \frac{1}{|\mathcal{V}|} \sum_{u,v \in \mathcal{V}}^{u \neq v} I\big(\mathbf{x}^{(u)}; \mathbf{z}^{(v)}|\mathbf{x}^{(v)}\big). \tag{3}$$

### 3.1.1 VIEW-SPECIFIC INFORMATION EXTRACTION AND CROSS-VIEW CONSISTENCY MODELING

In Eq. (3), the first term maximizes $I(\mathbf{z}^{(v)}; \mathbf{x}^{(v)})$ to preserve the view-specific information, and the second term minimizes $\sum_v I(\mathbf{x}^{(u)}; \mathbf{x}^{(v)} \mid \mathbf{z}^{(v)})$ to enforce cross-view consistency. We adopt the trade-off coefficient $\beta$ to balance information compactness and effectiveness.

$$\begin{aligned} I(\mathbf{x}^{(v)}; \mathbf{z}^{(v)}) &= \int \int p(\mathbf{x}^{(v)}, \mathbf{z}^{(v)}) \log \frac{p(\mathbf{x}^{(v)}|\mathbf{z}^{(v)})}{p(\mathbf{x}^{(v)})} d\mathbf{x}^{(v)} d\mathbf{z}^{(v)} \\ &\geq \int p(\mathbf{x}^{(v)}) \int p(\mathbf{z}^{(v)}|\mathbf{x}^{(v)}) \log p(\mathbf{x}^{(v)}|\mathbf{z}^{(v)}) d\mathbf{x}^{(v)} d\mathbf{z}^{(v)}. \end{aligned} \tag{4}$$

For the lower bound of Eq. (4), direct solution is intractable due to the unknown conditional distribution. Therefore we introduce a variational coding network $q^v(\mathbf{z}^{(v)} \mid \mathbf{x}^{(v)})$ to approximate $p(\mathbf{z}^{(v)} \mid \mathbf{x}^{(v)})$. Then Eq. (4) can be rewritten as follows:

$$\begin{aligned} I(\mathbf{x}^{(v)}; \mathbf{z}^{(v)}) &\geq \int p(\mathbf{x}^{(v)}) \int p(\mathbf{z}^{(v)}|\mathbf{x}^{(v)}) \log p(\mathbf{x}^{(v)}|\mathbf{z}^{(v)}) d\mathbf{x}^{(v)} d\mathbf{z}^{(v)} \\ &\geq \mathbb{E}_{\mathbf{x}^{(v)} \sim p(\mathbf{x}^{(v)})} \Big[ \int p(\mathbf{z}^{(v)}|\mathbf{x}^{(v)}) \log q^v(\mathbf{x}^{(v)}|\mathbf{z}^{(v)}) d\mathbf{z}^{(v)} \Big] \\ &= \mathbb{E}_{p(\mathbf{z}^{(v)}|\mathbf{x}^{(v)})} \Big[ \log q^v(\mathbf{x}^{(v)} \mid \mathbf{z}^{(v)}) \Big]. \end{aligned} \tag{5}$$

Obviously, we can maximize the lower bound in Eq. (5) to achieve our optimal goal of $\max I(\mathbf{x}^{(v)}; \mathbf{z}^{(v)})$. For the second term in Eq. (3), we have the following formulation:

$$\begin{aligned} &I(\mathbf{x}^{(u)}; \mathbf{z}^{(v)}|\mathbf{x}^{(v)}) \\ &= \int \int p(\mathbf{x}^{(u)}, \mathbf{x}^{(v)}, \mathbf{z}^{(v)}) \log \frac{p(\mathbf{x}^{(u)}, \mathbf{x}^{(v)}, \mathbf{z}^{(v)}) p(\mathbf{x}^{(v)})}{p(\mathbf{x}^{(u)}, \mathbf{x}^{(v)}) p(\mathbf{z}^{(v)}, \mathbf{x}^{(v)})} d\mathbf{x}^{(u)} d\mathbf{x}^{(v)} d\mathbf{z}^{(v)} \\ &= \int \int p(\mathbf{x}^{(u)}, \mathbf{x}^{(v)}, \mathbf{z}^{(v)}) \log \frac{p(\mathbf{z}^{(v)}|\mathbf{x}^{(u)}, \mathbf{x}^{(v)})}{p(\mathbf{z}^{(v)}|\mathbf{x}^{(v)})} d\mathbf{x}^{(u)} d\mathbf{x}^{(v)} d\mathbf{z}^{(v)}. \end{aligned} \tag{6}$$

Existing approaches typically employ a simple Multilayer Perceptron (MLP) to model the distribution of $p(\mathbf{z}^{(v)}|\mathbf{x}^{(v)})$. However, under our cross-view consistency encoding framework, where all $m$ views are expected to be encoded with shared semantic features, we decouple the process of approximating distribution $p(\mathbf{z}^{(v)}|\mathbf{x}^{(v)})$. Specifically, we model distributions $\{r_v^1(\mathbf{z}^{(1)}|\mathbf{x}^{(v)}),\ldots,r_v^m(\mathbf{z}^{(m)}|\mathbf{x}^{(v)})\}$ separately, which $\{r_v^n\}_{n=1}^m$ means the stochastic encoders from source view $v$ to target view $n$, and then employ PoE fusion to obtain distribution $r^v(\mathbf{z}^{(v)}|\mathbf{x}^{(v)})$ to approximate $p(\mathbf{z}^{(v)}|\mathbf{x}^{(v)})$:

$$p(\mathbf{z}^{(v)}|\mathbf{x}^{(v)}) \approx r^v(\mathbf{z}^{(v)}|\mathbf{x}^{(v)}) := r(\mathbf{z}^{(v)}) \prod_{n=1}^m r_v^n(\mathbf{z}^{(v)}|\mathbf{x}^{(n)}), \tag{7}$$

where we set $r(\mathbf{z}^{(v)})$ as a standard Gaussian distribution $r(\mathbf{z}^{(v)}) := \mathcal{N}(0, \mathbf{I})$ for a vanilla implementation. Then we can get the following variational upper bound:

$$\begin{aligned}
&I(\mathbf{x}^{(u)}; \mathbf{z}^{(v)}|\mathbf{x}^{(v)}) \\
&= \int \int p(\mathbf{x}^{(u)}, \mathbf{x}^{(v)}, \mathbf{z}^{(v)}) \log \frac{p(\mathbf{z}^{(v)}|\{\mathbf{x}^{(u)}, \mathbf{x}^{(v)}\})r^v(\mathbf{z}^{(v)}|\mathbf{x}^{(v)})}{p(\mathbf{z}^{(v)}|\mathbf{x}^{(v)})r^v(\mathbf{z}^{(v)}|\mathbf{x}^{(v)})} d\mathbf{x}^{(u)} d\mathbf{x}^{(v)} d\mathbf{z}^{(v)} \\
&\leq \int p(\mathbf{x}^{(u)}, \mathbf{x}^{(v)}, p(\mathbf{z}^{(v)}) \log \frac{p(\mathbf{z}^{(v)}|\mathbf{x}^{(u)}, \mathbf{x}^{(v)})}{r(\mathbf{z}^{(v)}|\mathbf{x}^{(u)})} d\mathbf{x}^{(u)} d\mathbf{x}^{(v)} d\mathbf{z}^{(v)} \\
&= \mathbb{E}_{\mathbf{x}^{(u)},\mathbf{x}^{(v)} \sim p(\mathbf{x}^{(u)},\mathbf{x}^{(v)})}[D_{KL}(p(\mathbf{z}^{(v)}|\mathbf{x}^{(v)})\|r^v(\mathbf{z}^{(v)}|\mathbf{x}^{(u)}))],
\end{aligned} \tag{8}$$

where $D_{KL}$ is the Kullback-Leibler divergence. Aggregating over Eqs. (5) and (8), and introducing the penalty coefficient $\beta$, we obtain the training objective for Eq. (3), i.e., minimizing the loss function $\mathcal{L}_{ib}$:

$$\begin{aligned}
\mathcal{L}_{ib} = \mathcal{L}_{re} + \beta\mathcal{L}_{pc} = &\frac{1}{|\mathcal{V}|} \sum_{v \in \mathcal{V}} \left[ -\mathbb{E}_{\mathbf{z}^{(v)} \sim p(\mathbf{z}^{(v)}|\mathbf{x}^{(v)})} \log q^v(\mathbf{x}^{(v)}|\mathbf{z}^{(v)}) \right] \\
&+ \beta \frac{1}{|\mathcal{V}|} \sum_{u,v \in \mathcal{V}}^{u \neq v} \sum_{n=1}^m D_{KL}(r_v^n(\mathbf{z}^{(v)}|\mathbf{x}^{(v)})\|r_u^n(\mathbf{z}^{(v)}|\mathbf{x}^{(u)})).
\end{aligned} \tag{9}$$

### 3.2 Prior alignment via random permutation

To regularize view-specific posteriors in $\mathcal{L}_{pc}$ while avoiding cubic complexity, we introduce a permutation-consistent alignment principle that serves a role analogous to cross-view prior sharing. From loss function $\mathcal{L}_{pc}$, it encourages the latent posterior of view $v$ to be close to that constructed from another view $u$. A naive implementation evaluates all ordered pairs $(u, v)$ with $u \neq v$, incurring $O(|\mathcal{V}|^3)$ complexity per batch. Instead, we attempt to randomize the association between views so that each view $v$ is matched with a single view $u \neq v$ per iteration, reducing the cost to $O(|\mathcal{V}|^2)$ while preserving the intended regularization.

**Definition 3.2** *For a given view $v$, the latent distribution cluster drawn on it is defined as:*

$$\mathcal{C}^v = \left\{ \mathbf{z}^{(v \to n)} \sim r_v^n(\mathbf{z}^{(n)} \mid \mathbf{x}^{(v)}) \right\}_{n=1}^m,$$

*where $\mathbf{z}^{(v \to n)}$ denotes the latent sub-variable from source view $v$ to target view $n$.*

**Proposition 3.3 (Permutation consistency)** *By randomly swapping the corresponding elements of the latent distribution clusters $\mathcal{C}^v$ and other available views, the distributions of the corresponding latent sub-variables within each cluster remain consistent before and after the swap, i.e., let $\pi = \{\pi_i\}_{i=1}^m, \pi_i \in \mathcal{V}$ be a random view index sequence of length $m$, and $\widetilde{\mathcal{C}}^v = \left\{ \mathbf{z}^{(\pi_n \to n)} \sim r_{\pi_n}^n(\mathbf{z}^{(n)} \mid \mathbf{x}^{(\pi_n)}) \right\}_{n=1}^m$ be the cluster after random permutation, then we have:*

$$D_{KL}\left(\mathbf{z}^{(v \to n)} \,\middle\|\, \mathbf{z}^{(\pi_n \to n)}\right) = 0, \qquad \forall\, n \in \{1, \ldots, m\}.$$

The permutation consistency defined in the proposition is based on the assumption of semantic encoding consistency in the latent space, i.e., the ideal cross-view representations should contain only

shared semantic information. This design enforces the network to encode semantic information from all available views, ensuring cross-view semantic alignment while effectively improving computational parallelism. Note that, given the diversity of source views within each cluster, we further constrain the selection of $\pi$ such that $|\{\pi_i\}_{i=1}^m| = |\mathcal{V}|$; that is, we randomly select view indices from the available view set $\mathcal{V}$ without replacement. For example, if $\mathcal{V} = \{1, 2, 4, 6\}$, a possible permutation of the indices $\pi$ could be $\{4, 1, 6, 2\}$.

### 3.3 INTRA- AND INTER-VIEW ALIGNMENT

To further enhance cross-view semantic coherence and robustness within the PCVE framework, we introduce two complementary alignment objectives: an intra-view alignment term that encourages each view-specific latent posterior to match its corresponding PoE-fused counterpart, and an inter-view alignment term that aggregates per-view means across views in a multi-view fashion.

**Intra-view alignment.** Let $r_v^n(\mathbf{z}^{(v)} \mid \mathbf{x}^{(v)})$ denote the stochastic encoder that maps source view $v$ to the target latent variable of view $n$, and let $r^v(\mathbf{z}^{(v)} \mid \mathbf{x}^{(v)})$ be the view-$v$ posterior obtained by PoE fusion over the set of available views $\mathcal{V}$. For each sample, we define an intra-view KL divergence that encourages the per-source encoders to align with the fused posterior on the same latent component:

$$\mathcal{L}_{intra} = \frac{1}{|\mathcal{V}|} \sum_{n=1}^m \sum_{v \in \mathcal{V}} D_{KL}(r_v^n(\mathbf{z}^{(v)} \mid \mathbf{x}^{(v)}) | r^v(\mathbf{z}^{(v)} \mid \mathbf{x}^{(v)})). \tag{10}$$

Intuitively, this term drives the latent distribution cluster $\mathcal{C}^v$ induced by view $v$ to align with its PoE-fused posterior, preventing view-specific drift and reinforcing distributional consistency at the component level.

**Inter-view alignment.** Besides the intra-view alignment, we also encourage cross-view consensus by directly regularizing the per-view posteriors toward a shared latent mean. Specifically, we adopt a softmax-weighted similarity matching across views of the same instance. Let $\mu_i^{(v)} \in \mathbb{R}^d$ be the embedding of sample $i$ under view $v$, and let $\tilde{\mu}_i^{(v)} = \mu_i^{(v)} / \|\mu_i^{(v)}\|_2$ denote its $\ell_2$-normalized form and $\mathcal{V}_i$ denote the available view set of sample $i$. Define the temperature-scaled cosine similarity $s_{i,j}^{(v,u)} = \frac{\langle \tilde{\mu}_i^{(v)}, \tilde{\mu}_j^{(u)} \rangle}{\tau}$, where $\tau > 0$ is a temperature. The inter-view alignment loss is the negative log-likelihood of assigning the anchor to its within-instance views:

$$\mathcal{L}_{inter} = -\sum_i \sum_{u,v \in \mathcal{V}_i}^{u \neq v} \log \frac{\exp(s_{i,i}^{(v,u)})}{\sum_j \exp(s_{i,j}^{(v,u)})}. \tag{11}$$

This objective increases the probability mass on same-instance cross-view pairs and reduces it on different-instance pairs, which naturally accommodates missing views by restricting sums to the available view sets.

### 3.4 MULTI-LABEL CLASSIFICATION AND OVERALL OBJECTIVE

After obtaining the components of latent shared representation on each view, we fuse available views via a PoE to form the joint posterior: $q(\mathbf{z} \mid \{\mathbf{x}^{(v)}\}_{v \in \mathcal{V}}) \propto r(z) \prod_{v \in \mathcal{V}} r^v(\mathbf{z}^{(v)} \mid \mathbf{x}^{(v)})$. To enable a differentiable prediction path, we sample latent variable from $q(\mathbf{z} \mid \{\mathbf{x}^{(v)}\}_{v \in \mathcal{V}})$ using the reparameterization trick. Let $q(\mathbf{z} \mid \{\mathbf{x}^{(v)}\}_{v \in \mathcal{V}}) = \mathcal{N}(\mu_{\text{poe}}, \Sigma_{\text{poe}})$ with mean $\mu_{\text{poe}}$ and covariance $\Sigma_{\text{poe}}$, a $D$ times sample is: $\bar{z} = \frac{1}{D} \sum_{d=1}^D \mu_{\text{poe}} + \Sigma_{\text{poe}}^{1/2} \odot \epsilon^{(d)}$, $\epsilon^{(d)} \sim \mathcal{N}(0, I)$. Then $\bar{z}$ is mapped to multi-label prediction probabilities $p \in [0, 1]^C$ via a small MLP followed by a Sigmoid layer: $p = \sigma(f_c(\bar{z}))$. In the missing-label setting, supervision is accumulated only over the available label set $\mathcal{G} \subseteq \{1, \ldots, C\}$. The multi-label cross-entropy loss is:

$$\mathcal{L}_{ce} = -\frac{1}{|\mathcal{G}|} \sum_{i \in \mathcal{G}} \Big[ y_i \log p_i + (1 - y_i) \log(1 - p_i) \Big], \tag{12}$$

where $y_i \in \{0, 1\}$ is the observed label for class $i$ and $p_i$ is the predicted probability and unknown labels ($i \notin \mathcal{G}$) are excluded. Aggregating the regularization and task terms, our overall objective comprises permutation consistency loss $\mathcal{L}_{pc}$, reconstruction loss, missing multi-label classification loss $\mathcal{L}_{ce}$, and intra/inter-view alignment loss $\mathcal{L}_{\text{inter}} + \mathcal{L}_{\text{intra}}$:

$$\mathcal{L} = \mathcal{L}_{ce} + \alpha \, \mathcal{L}_{ib} + \gamma \, \mathcal{L}_{inter} + \lambda \, \mathcal{L}_{intra} \tag{13}$$

where $\alpha \geq 0$ balance the proportion of IB strategy. Note that the $\mathcal{L}_{ib}$ is composed of $\mathcal{L}_{pc}$ and $\mathcal{L}_{re}$ with coefficient $\beta$ to balance the information compression and retention across views. Penalty parameters $\gamma$ and $\lambda$ are set to $0.1$ for convenience.

## 4 EXPERIMENTS

We introduce the experimental setup, main experimental results and analysis in this section. For other experimental results and ablation study, please refer to the appendix.

### 4.1 EXPERIMENTAL SETTINGS

**Datasets.** Following common practice (Liu et al., 2023a; Li & Chen, 2023), we evaluate on five standard multi-view multi-label benchmarks: Corel5k Duygulu et al. (2002), Pascal07 Everingham et al. (2009), ESPGame Von Ahn & Dabbish (2004), IAPRTC12 Henning et al. (2006), and MIR-FLICKR Huiskes & Lew (2008). Each sample provides six views (GIST, HSV, DenseHue, DenseSift, RGB, LAB). We adopt the same data statistics and feature settings as prior work to ensure fair comparison.

**Incomplete data construction and splits.** To simulate the doubly-incomplete setting, we randomly mask views and labels: for each sample, we drop the view with a fixed probability while ensuring at least one view remains; for labels, we randomly mask both positive and negative entries with the same probability. Unless otherwise specified, the missing-view rate and missing-label rate are both set to 50%. Then, we split all data into [70%/15%/15%] for training, validation, and test.

**Baselines.** We compare our PCVE against nine strong methods: complete multi-view multi-label learning method like CDMM (Zhao et al., 2021) and LVSL (Zhao et al., 2022); single-view missing multi-label methods DM2L (Ma & Chen, 2021); and methods tailored for iM3C, like iMVWL (Tan et al., 2018), NAIM3L (Li & Chen, 2023), DICNet (Liu et al., 2023b), DIMC (Wen et al., 2023), MSLPP (Long et al., 2024) and SIP (Liu et al., 2024b). For methods not natively supporting iM3C task, we follow standard adaptations in previous work (Zhao et al., 2022): mean imputation over available views for methods unsuitable for missing views, and reporting the best single-view for single-view method; for methods not supporting missing labels, unknown entries are excluded from the supervision term. We use authors' code and recommended hyperparameters when available.

**Metrics.** We report six widely-used metrics: Ranking Loss (RL), Average Precision (AP), Hamming Loss (HL), Area Under ROC Curve (AUC), One-Error (OE), and Coverage (Cov). To unify the direction, we present $1-$RL, $1-$HL, $1-$OE, $1-$Cov along with AP and AUC, so higher is better for all. Each experiment is repeated multiple times with reported mean and standard deviation.

**Implementation details.** For our method PCVE, the latent dimension is 512, batch size 128, optimizer SGD with initial learning rate 0.001. Per mini-batch we draw 10 samples for latent variables and use their average as a robust estimate. Training is conducted on Ubuntu with a single NVIDIA RTX 5090 GPU under PyTorch 2.x. The key hyperparameters are selected via grid search on the validation set and fixed for test reporting.

### 4.2 EXPERIMENTAL RESULTS AND ANALYSIS

We compare PCVE with the nine baselines on all five datasets under 50% missing views and 50% missing labels. From Table 1, we can observe that our PCVE matches or outperforms the top baseline across all six metrics on all datasets. Compared with the best prior baselines, PCVE exhibits statistically consistent improvements. For example: On the Corel5k and Pascal07, PCVE improves

Table 1: Results under 50% missing views and 50% missing labels. The decimal in the lower right corner is the standard deviation. The best result is marked in **bold**.

| Data | Metric | CDMM | DM2L | LVSL | iMVWL | NAIM3L | DICNet | DIMC | MSLPP | SIP | **PCVE** |
|---|---|---|---|---|---|---|---|---|---|---|---|
| Corel5k | AP ↑ | $0.354_{0.004}$ | $0.262_{0.005}$ | $0.342_{0.004}$ | $0.283_{0.008}$ | $0.309_{0.004}$ | $0.381_{0.004}$ | $0.353_{0.006}$ | $0.413_{0.008}$ | $0.418_{0.009}$ | $\mathbf{0.421}_{0.008}$ |
| | 1-HL ↑ | $0.987_{0.000}$ | $0.987_{0.000}$ | $0.987_{0.000}$ | $0.978_{0.000}$ | $0.987_{0.000}$ | $0.988_{0.000}$ | $0.987_{0.000}$ | $0.988_{0.000}$ | $0.988_{0.000}$ | $\mathbf{0.988}_{0.000}$ |
| | 1-RL ↑ | $0.884_{0.003}$ | $0.843_{0.002}$ | $0.881_{0.003}$ | $0.865_{0.005}$ | $0.878_{0.002}$ | $0.882_{0.004}$ | $0.867_{0.001}$ | $0.901_{0.003}$ | $\mathbf{0.911}_{0.003}$ | $0.910_{0.003}$ |
| | AUC ↑ | $0.888_{0.003}$ | $0.845_{0.002}$ | $0.884_{0.003}$ | $0.868_{0.005}$ | $0.881_{0.002}$ | $0.884_{0.004}$ | $0.870_{0.001}$ | $0.903_{0.004}$ | $0.913_{0.003}$ | $\mathbf{0.913}_{0.002}$ |
| | 1-OE ↑ | $0.410_{0.007}$ | $0.295_{0.014}$ | $0.391_{0.009}$ | $0.311_{0.015}$ | $0.350_{0.009}$ | $0.468_{0.007}$ | $0.422_{0.015}$ | $0.485_{0.010}$ | $0.489_{0.016}$ | $\mathbf{0.493}_{0.013}$ |
| | 1-Cov ↑ | $0.723_{0.007}$ | $0.647_{0.005}$ | $0.718_{0.006}$ | $0.702_{0.008}$ | $0.725_{0.005}$ | $0.727_{0.011}$ | $0.684_{0.011}$ | $0.766_{0.009}$ | $0.787_{0.009}$ | $\mathbf{0.790}_{0.009}$ |
| Pascal07 | AP ↑ | $0.508_{0.005}$ | $0.471_{0.008}$ | $0.504_{0.005}$ | $0.437_{0.018}$ | $0.488_{0.003}$ | $0.505_{0.012}$ | $0.532_{0.002}$ | $0.544_{0.010}$ | $0.555_{0.010}$ | $\mathbf{0.559}_{0.008}$ |
| | 1-HL ↑ | $0.931_{0.001}$ | $0.928_{0.001}$ | $0.930_{0.000}$ | $0.882_{0.004}$ | $0.928_{0.001}$ | $0.929_{0.001}$ | $0.931_{0.001}$ | $0.932_{0.001}$ | $0.931_{0.001}$ | $\mathbf{0.934}_{0.001}$ |
| | 1-RL ↑ | $0.812_{0.004}$ | $0.761_{0.005}$ | $0.806_{0.003}$ | $0.736_{0.015}$ | $0.783_{0.001}$ | $0.783_{0.008}$ | $0.813_{0.000}$ | $0.819_{0.006}$ | $0.830_{0.004}$ | $\mathbf{0.834}_{0.004}$ |
| | AUC ↑ | $0.838_{0.003}$ | $0.779_{0.004}$ | $0.832_{0.002}$ | $0.767_{0.015}$ | $0.811_{0.001}$ | $0.809_{0.006}$ | $0.833_{0.002}$ | $0.841_{0.004}$ | $0.850_{0.005}$ | $\mathbf{0.857}_{0.004}$ |
| | 1-OE ↑ | $0.419_{0.008}$ | $0.420_{0.011}$ | $0.419_{0.006}$ | $0.362_{0.024}$ | $0.421_{0.006}$ | $0.427_{0.015}$ | $0.456_{0.011}$ | $0.466_{0.014}$ | $0.464_{0.018}$ | $\mathbf{0.471}_{0.013}$ |
| | 1-Cov ↑ | $0.759_{0.003}$ | $0.692_{0.004}$ | $0.751_{0.003}$ | $0.677_{0.015}$ | $0.727_{0.002}$ | $0.731_{0.006}$ | $0.769_{0.007}$ | $0.771_{0.003}$ | $0.783_{0.006}$ | $\mathbf{0.790}_{0.006}$ |
| ESPGame | AP ↑ | $0.289_{0.003}$ | $0.212_{0.002}$ | $0.285_{0.003}$ | $0.244_{0.005}$ | $0.246_{0.002}$ | $0.297_{0.002}$ | $0.287_{0.002}$ | $0.310_{0.003}$ | $0.311_{0.004}$ | $\mathbf{0.314}_{0.004}$ |
| | 1-HL ↑ | $0.983_{0.000}$ | $0.982_{0.000}$ | $0.983_{0.000}$ | $0.972_{0.000}$ | $0.983_{0.000}$ | $0.983_{0.000}$ | $0.983_{0.000}$ | $0.983_{0.000}$ | $0.983_{0.000}$ | $\mathbf{0.983}_{0.000}$ |
| | 1-RL ↑ | $0.832_{0.001}$ | $0.781_{0.001}$ | $0.829_{0.001}$ | $0.808_{0.002}$ | $0.818_{0.002}$ | $0.832_{0.001}$ | $0.821_{0.000}$ | $0.843_{0.002}$ | $0.849_{0.002}$ | $\mathbf{0.852}_{0.002}$ |
| | AUC ↑ | $0.836_{0.001}$ | $0.785_{0.001}$ | $0.833_{0.002}$ | $0.813_{0.002}$ | $0.824_{0.002}$ | $0.836_{0.001}$ | $0.826_{0.000}$ | $0.847_{0.002}$ | $0.853_{0.002}$ | $\mathbf{0.856}_{0.002}$ |
| | 1-OE ↑ | $0.396_{0.005}$ | $0.294_{0.006}$ | $0.389_{0.004}$ | $0.343_{0.013}$ | $0.339_{0.003}$ | $0.439_{0.007}$ | $0.435_{0.009}$ | $0.457_{0.012}$ | $0.455_{0.007}$ | $\mathbf{0.460}_{0.008}$ |
| | 1-Cov ↑ | $0.574_{0.004}$ | $0.488_{0.003}$ | $0.567_{0.005}$ | $0.548_{0.004}$ | $0.571_{0.003}$ | $0.593_{0.003}$ | $0.562_{0.004}$ | $0.622_{0.005}$ | $0.628_{0.005}$ | $\mathbf{0.634}_{0.005}$ |
| IAPRTC12 | AP ↑ | $0.305_{0.004}$ | $0.234_{0.003}$ | $0.304_{0.004}$ | $0.237_{0.003}$ | $0.261_{0.001}$ | $0.323_{0.001}$ | $0.308_{0.001}$ | $\mathbf{0.340}_{0.005}$ | $0.331_{0.006}$ | $0.336_{0.005}$ |
| | 1-HL ↑ | $0.981_{0.000}$ | $0.980_{0.000}$ | $0.981_{0.000}$ | $0.969_{0.000}$ | $0.980_{0.000}$ | $0.981_{0.000}$ | $0.980_{0.000}$ | $0.981_{0.000}$ | $0.980_{0.000}$ | $\mathbf{0.981}_{0.000}$ |
| | 1-RL ↑ | $0.862_{0.002}$ | $0.823_{0.002}$ | $0.861_{0.002}$ | $0.833_{0.002}$ | $0.848_{0.001}$ | $0.873_{0.001}$ | $0.864_{0.000}$ | $0.882_{0.002}$ | $0.885_{0.003}$ | $\mathbf{0.888}_{0.003}$ |
| | AUC ↑ | $0.864_{0.002}$ | $0.825_{0.001}$ | $0.863_{0.001}$ | $0.835_{0.001}$ | $0.850_{0.001}$ | $0.874_{0.000}$ | $0.864_{0.000}$ | $0.883_{0.002}$ | $0.886_{0.002}$ | $\mathbf{0.889}_{0.002}$ |
| | 1-OE ↑ | $0.432_{0.008}$ | $0.340_{0.006}$ | $0.429_{0.009}$ | $0.352_{0.008}$ | $0.390_{0.005}$ | $0.468_{0.002}$ | $0.431_{0.006}$ | $0.474_{0.008}$ | $0.463_{0.009}$ | $\mathbf{0.477}_{0.007}$ |
| | 1-Cov ↑ | $0.597_{0.004}$ | $0.529_{0.004}$ | $0.597_{0.004}$ | $0.564_{0.005}$ | $0.592_{0.004}$ | $0.649_{0.001}$ | $0.597_{0.004}$ | $0.672_{0.006}$ | $0.675_{0.007}$ | $\mathbf{0.680}_{0.006}$ |
| MIRFLICKR | AP ↑ | $0.570_{0.002}$ | $0.514_{0.006}$ | $0.553_{0.002}$ | $0.490_{0.012}$ | $0.551_{0.002}$ | $0.589_{0.005}$ | $0.602_{0.002}$ | $0.615_{0.004}$ | $0.614_{0.004}$ | $\mathbf{0.618}_{0.004}$ |
| | 1-HL ↑ | $0.886_{0.001}$ | $0.878_{0.001}$ | $0.885_{0.001}$ | $0.839_{0.002}$ | $0.882_{0.001}$ | $0.888_{0.002}$ | $0.888_{0.000}$ | $0.892_{0.001}$ | $0.891_{0.001}$ | $\mathbf{0.895}_{0.001}$ |
| | 1-RL ↑ | $0.856_{0.001}$ | $0.831_{0.003}$ | $0.856_{0.001}$ | $0.803_{0.008}$ | $0.844_{0.001}$ | $0.863_{0.004}$ | $0.865_{0.001}$ | $0.879_{0.002}$ | $0.877_{0.002}$ | $\mathbf{0.880}_{0.002}$ |
| | AUC ↑ | $0.846_{0.001}$ | $0.828_{0.003}$ | $0.844_{0.001}$ | $0.787_{0.012}$ | $0.837_{0.001}$ | $0.849_{0.004}$ | $0.852_{0.001}$ | $0.865_{0.002}$ | $0.860_{0.003}$ | $\mathbf{0.868}_{0.002}$ |
| | 1-OE ↑ | $0.631_{0.004}$ | $0.510_{0.008}$ | $0.607_{0.004}$ | $0.511_{0.022}$ | $0.585_{0.003}$ | $0.637_{0.007}$ | $0.647_{0.007}$ | $0.667_{0.007}$ | $0.662_{0.008}$ | $\mathbf{0.670}_{0.006}$ |
| | 1-Cov ↑ | $0.640_{0.001}$ | $0.604_{0.005}$ | $0.636_{0.001}$ | $0.572_{0.013}$ | $0.631_{0.002}$ | $0.652_{0.007}$ | $0.661_{0.003}$ | $0.679_{0.003}$ | $0.678_{0.003}$ | $\mathbf{0.682}_{0.003}$ |

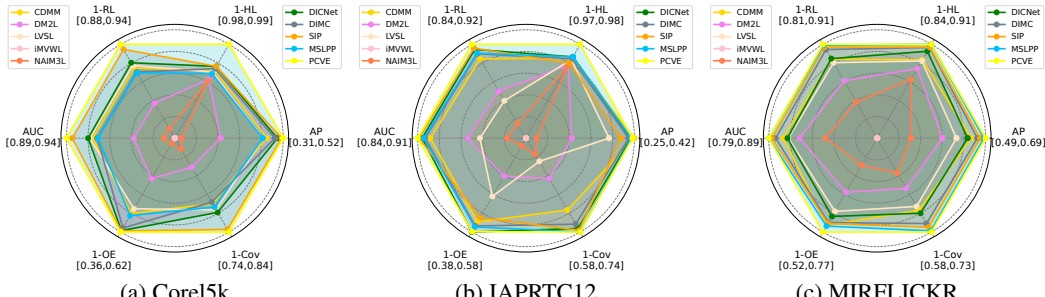

(a) Corel5k      (b) IAPRTC12      (c) MIRFLICKR

Figure 2: Experimental results of ten methods on the three full databases without any missing views or labels. The center of the radar map shows the worst results and the vertexes mean the best results on the six metrics.

AP and AUC by about 0.3–0.7 points over the second best method SIP; On the larger-scale datasets with around 20k samples like ESPGame, IAPRTC12, and MIRFLICKR, the improvement trends are modest yet steady across all six metrics, indicating improved robustness under severe double-incompleteness. In addition, a holistic view of Table 1 shows that methods explicitly designed for the iM3C setting (e.g., DICNet, SIP, and our PCVE) systematically outperform those not natively addressing dual incompleteness (e.g., CDMM and LVSL). For example, on Pascal07, iM3C-oriented methods (SIP with AP 0.555; PCVE with AP 0.559) surpass non-iM3C counterparts (e.g., CDMM with AP 0.508 and LVSL with AP 0.504). Similar trends recur on other datasets, which underscores the necessity of introducing prior missing information, rather than relying on coarse imputations.

We further evaluate all methods under the complete data setting by setting both the view-missing and label-missing rates to 0. The radar plots in Fig. 2 visualize results on three representative datasets, where curves closer to the center indicate worse performance. Two observations stand out: Incompleteness imposes a substantial negative impact on all methods; Even with full views and labels, PCVE maintains a consistent edge over strong baselines, confirming its compatibility

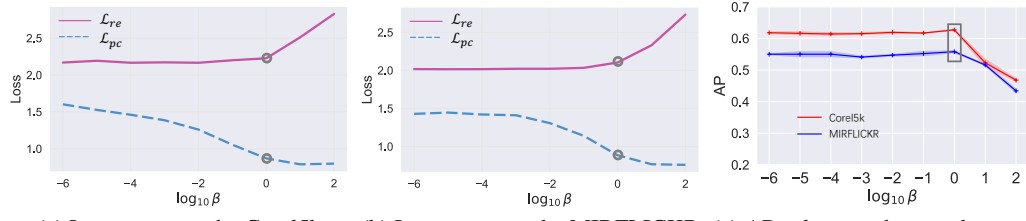

(a) Loss curve on the Corel5k    (b) Loss curve on the MIRFLICKR  (c) AP values on the two datasets

Figure 3: Impact of different information-balance parameter $\beta$ on loss $\mathcal{L}_{re}$, $\mathcal{L}_{pc}$ (3a,3b), and AP (3c). The blue area shows the standard deviation.

beyond the doubly-incomplete regime. In particular, on the Corel5k dateset, PCVE achieves notable advantages on 1-HL and 1-Cov compared to the second-best approach.

### 4.3 ANALYSIS OF THE BALANCE IN THE IB.

We further examine how PCVE balances representation sufficiency and compactness through its IB loss by tuning the trade-off between the reconstruction term $\mathcal{L}_{re}$ and the cross-view permutation consistency term $\mathcal{L}_{pc}$ in Eq. (9), where $\mathcal{L}_{re}$ aims to preserve view-specific information necessary for reconstruction, while $\mathcal{L}_{pc}$ for enforcing semantic invariance across views by aligning their posteriors toward a compact task-relevant shared space. To be specific, we sweep $\beta$ on Corel5k and Pascal07 under the doubly-incomplete setting and track the dynamics of $\mathcal{L}_{re}$, $\mathcal{L}_{pc}$, and AP values across training (see Fig. 3). Three consistent phenomena emerge: (i) When $\beta$ is too small, $\mathcal{L}_{re}$ dominates and the model over-retains view-specific information, leading to weaker cross-view generalization and degraded performance despite low reconstruction error. (ii) When $\beta$ is too large, excessive compression harms sufficiency as well: $\mathcal{L}_{pc}$ remains low but $\mathcal{L}_{re}$ rises rapidly, accompanied by noticeable declines in AP, indicating information underflow. (iii) A moderate balance yields the best outcomes: performance peaks when the pull between $\mathcal{L}_{re}$ and $\mathcal{L}_{pc}$ reaches a balance (typically $\beta$ in the mid-range), suggesting that PCVE benefits from jointly promoting semantic alignment and preserving reconstructive fidelity.

## 5 CONCLUSION

In this paper, we present PCVE, a permutation-consistent variational encoding framework for incomplete multi-view multi-label classification. We adopt an information bottleneck framework that couples view-specific information preservation with a permutation-based cross-view consistency objective to achieve efficient semantic alignment across views. Our key contribution is an early-stage consistency regularization based on distribution-cluster swapping, which effectively suppresses view redundancy and promotes the learning of sufficient shared representations. Extensive experiments on five benchmarks under both doubly-incomplete and fully observed settings show consistent gains across all metrics. Analyses further verify that a balanced trade-off between information preservation and compression is critical to avoid information collapse and redundancy. PCVE offers a general and effective solution for iM3C task, with potential to extend to broader multi-view learning scenarios.

### ACKNOWLEDGMENTS

This work was supported by the Science and Technology Development Fund (FDCT), Macao S.A.R under Grant 0028/2023/RIA1, and the National Natural Science Foundation of China under Grant 62372136.

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

# A  APPENDIX

## A.1  USE OF LARGE LANGUAGE MODELS (LLMS)

We used large language model solely as a writing assistant to improve the clarity, grammar, and style of the manuscript. The model was not involved in research ideation, experimental design, implementation, analysis, or result interpretation. We sincerely appreciate the contribution of the large language model in enhancing the readability and linguistic quality of this work. Its assistance was instrumental in refining the presentation of our research. All technical content, including methods, experiments, and conclusions, was fully developed and verified by the authors. The authors take full responsibility for the content of this paper.

## A.2  COMPLETE DERIVATION OF SHARED INFORMATION LEARNING MODEL

In this section, we give a detailed derivation of model (3):

$$\max \ \frac{1}{|\mathcal{V}|} \sum_{v \in \mathcal{V}} I\big(\mathbf{z}^{(v)}; \mathbf{x}^{(v)}\big) \ - \ \beta \cdot \frac{1}{|\mathcal{V}|} \sum_{u,v \in \mathcal{V}}^{u \neq v} I\big(\mathbf{x}^{(u)}; \mathbf{z}^{(v)} | \mathbf{x}^{(v)}\big). \tag{14}$$

For the first term in Eq. (14), we have:

$$
\begin{aligned}
& I(\mathbf{x}^{(v)}; \mathbf{z}^{(v)}) \\
= & \int \int p(\mathbf{x}^{(v)}, \mathbf{z}^{(v)}) \log \frac{p(\mathbf{x}^{(v)}|\mathbf{z}^{(v)})}{p(\mathbf{x}^{(v)})} d\mathbf{x}^{(v)} d\mathbf{z}^{(v)} \\
= & \Big[ \int \int p(\mathbf{x}^{(v)}, \mathbf{z}^{(v)}) \log p(\mathbf{x}^{(v)}|\mathbf{z}^{(v)}) d\mathbf{x}^{(v)} d\mathbf{z}^{(v)} + \\
& \int p(\mathbf{z}^{(v)}|\mathbf{x}^{(v)}) \int p(\mathbf{x}^{(v)}) \log \frac{1}{p(\mathbf{x}^{(v)})} d\mathbf{x}^{(v)} d\mathbf{z}^{(v)} \Big] \\
= & \Big[ \int \int p(\mathbf{x}^{(v)}, \mathbf{z}^{(v)}) \log p(\mathbf{x}^{(v)}|\mathbf{z}^{(v)}) d\mathbf{x}^{(v)} d\mathbf{z}^{(v)} + H(\mathbf{x}^{(v)}) \Big]
\end{aligned}
\tag{15}
$$

Due to the information entropy $H(\mathbf{x}^{(v)}) \geq 0$, we have:

$$
\begin{aligned}
& I(\mathbf{x}^{(v)}; \mathbf{z}^{(v)}) \\
\geq & \int \int p(\mathbf{x}^{(v)}, \mathbf{z}^{(v)}) \log p(\mathbf{x}^{(v)}|\mathbf{z}^{(v)}) d\mathbf{x}^{(v)} d\mathbf{z}^{(v)} \\
= & \int p(\mathbf{x}^{(v)}) \int p(\mathbf{z}^{(v)}|\mathbf{x}^{(v)}) \log q^v(\mathbf{x}^{(v)}|\mathbf{z}^{(v)}) d\mathbf{x}^{(v)} d\mathbf{z}^{(v)} + \\
& \int p(\mathbf{z}^{(v)}) \int p(\mathbf{x}^{(v)}|\mathbf{z}^{(v)}) \log \frac{p(\mathbf{x}^{(v)}|\mathbf{z}^{(v)})}{q^v(\mathbf{x}^{(v)}|\mathbf{z}^{(v)})} d\mathbf{x}^{(v)} d\mathbf{z}^{(v)} \\
= & \int p(\mathbf{x}^{(v)}) \int p(\mathbf{z}^{(v)}|\mathbf{x}^{(v)}) \log q^v(\mathbf{x}^{(v)}|\mathbf{z}^{(v)}) d\mathbf{x}^{(v)} d\mathbf{z}^{(v)} + \\
& \int p(\mathbf{z}^{(v)}) D_{KL}(p(\mathbf{x}^{(v)}|\mathbf{z}^{(v)}) \| q^v(\mathbf{x}^{(v)}|\mathbf{z}^{(v)})) d\mathbf{x}^{(v)} d\mathbf{z}^{(v)}
\end{aligned}
\tag{16}
$$

Since $D_{KL}(p(\mathbf{x}^{(v)}|\mathbf{z}^{(v)}) \| q^v(\mathbf{x}^{(v)}|\mathbf{z}^{(v)})) \geq 0$, we can get:

$$
\begin{aligned}
& I(\mathbf{x}^{(v)}; \mathbf{z}^{(v)}) \\
\geq & \int p(\mathbf{x}^{(v)}) \int p(\mathbf{z}^{(v)}|\mathbf{x}^{(v)}) \log q^v(\mathbf{x}^{(v)}|\mathbf{z}^{(v)}) d\mathbf{x}^{(v)} d\mathbf{z}^{(v)} \\
= & \int \int p(\mathbf{x}^{(v)}, \mathbf{z}^{(v)}) \log q^v(\mathbf{x}^{(v)}|\mathbf{z}^{(v)}) d\mathbf{x}^{(v)} d\mathbf{z}^{(v)} \\
= & \int p(\mathbf{x}^{(v)}) \int p(\mathbf{z}^{(v)}|\mathbf{x}^{(v)}) \log q^v(\mathbf{x}^{(v)}|\mathbf{z}^{(v)}) d\mathbf{x}^{(v)} d\mathbf{z}^{(v)} \\
= & \mathbb{E}_{p(\mathbf{z}^{(v)}|\mathbf{x}^{(v)})} \big[ \log q^v(\mathbf{x}^{(v)} \mid \mathbf{z}^{(v)}) \big].
\end{aligned}
\tag{17}
$$

For the latter term in Model (14), we have:

$$
\begin{aligned}
& I(\mathbf{x}^{(u)}; \mathbf{z}^{(v)}|\mathbf{x}^{(v)}) \\
= & \int \int p(\mathbf{x}^{(u)}, \mathbf{x}^{(v)}, \mathbf{z}^{(v)}) \log \frac{p(\mathbf{x}^{(u)}, \mathbf{x}^{(v)}, \mathbf{z}^{(v)}) p(\mathbf{x}^{(v)})}{p(\mathbf{x}^{(u)}, \mathbf{x}^{(v)}) p(\mathbf{z}^{(v)}, \mathbf{x}^{(v)})} d\mathbf{x}^{(u)} d\mathbf{x}^{(v)} d\mathbf{z}^{(v)} \\
= & \int \int p(\mathbf{x}^{(u)}, \mathbf{x}^{(v)}, \mathbf{z}^{(v)}) \log \frac{p(\mathbf{z}^{(v)}|\mathbf{x}^{(u)}, \mathbf{x}^{(v)})}{p(\mathbf{z}^{(v)}|\mathbf{x}^{(v)})} d\mathbf{x}^{(u)} d\mathbf{x}^{(v)} d\mathbf{z}^{(v)}.
\end{aligned}
\tag{18}
$$

By introducing approximate distribution $r^v(\mathbf{z}^{(v)}|\mathbf{x}^{(v)})$, we have:

$$
\begin{aligned}
& I(\mathbf{x}^{(u)}; \mathbf{z}^{(v)}|\mathbf{x}^{(v)}) \\
&= \int\int p(\mathbf{x}^{(u)}, \mathbf{x}^{(v)}, \mathbf{z}^{(v)}) \log \frac{p(\mathbf{z}^{(v)}|\mathbf{x}^{(u)}, \mathbf{x}^{(v)}) r^v(\mathbf{z}^{(v)}|\mathbf{x}^{(v)})}{p(\mathbf{z}^{(v)}|\mathbf{x}^{(v)}) r^v(\mathbf{z}^{(v)}|\mathbf{x}^{(v)})} d\mathbf{x}^{(u)} d\mathbf{x}^{(v)} d\mathbf{z}^{(v)} \\
&= \int\int p(\mathbf{x}^{(u)}, \mathbf{x}^{(v)}, \mathbf{z}^{(v)}) \log \frac{p(\mathbf{z}^{(v)}|\mathbf{x}^{(u)}, \mathbf{x}^{(v)})}{r^v(\mathbf{z}^{(v)}|\mathbf{x}^{(v)})} d\mathbf{x}^{(u)} d\mathbf{x}^{(v)} d\mathbf{z}^{(v)} + \\
&\quad \int\int p(\mathbf{x}^{(u)}, \mathbf{x}^{(v)}, \mathbf{z}^{(v)}) \log \frac{r^v(\mathbf{z}^{(v)}|\mathbf{x}^{(v)})}{p(\mathbf{z}^{(v)}|\mathbf{x}^{(v)})} d\mathbf{x}^{(u)} d\mathbf{x}^{(v)} d\mathbf{z}^{(v)} \\
&= \int\int p(\mathbf{x}^{(u)}, \mathbf{x}^{(v)}, \mathbf{z}^{(v)}) \log \frac{p(\mathbf{z}^{(v)}|\mathbf{x}^{(u)}, \mathbf{x}^{(v)})}{r^v(\mathbf{z}^{(v)}|\mathbf{x}^{(v)})} d\mathbf{x}^{(u)} d\mathbf{x}^{(v)} d\mathbf{z}^{(v)} + \\
&\quad \int\int p(\mathbf{x}^{(u)}, \mathbf{x}^{(v)})[\int p(\mathbf{z}^{(v)}|\mathbf{x}^{(v)}) \log \frac{r^v(\mathbf{z}^{(v)}|\mathbf{x}^{(v)})}{p(\mathbf{z}^{(v)}|\mathbf{x}^{(v)}))}] d\mathbf{z}^{(v)} d\mathbf{x}^{(u)} d\mathbf{x}^{(v)} \\
&= \int\int p(\mathbf{x}^{(u)}, \mathbf{x}^{(v)}, \mathbf{z}^{(v)}) \log \frac{p(\mathbf{z}^{(v)}|\mathbf{x}^{(u)}, \mathbf{x}^{(v)})}{r^v(\mathbf{z}^{(v)}|\mathbf{x}^{(v)})} d\mathbf{x}^{(u)} d\mathbf{x}^{(v)} d\mathbf{z}^{(v)} - \\
&\quad \int\int p(\mathbf{x}^{(u)}, \mathbf{x}^{(v)})[D_{KL}(r^v(\mathbf{z}^{(v)}|\mathbf{x}^{(v)})||p(\mathbf{z}^{(v)}|\mathbf{x}^{(v)})))] d\mathbf{z}^{(v)} d\mathbf{x}^{(u)} d\mathbf{x}^{(v)} \\
&\le \int p(\mathbf{x}^{(u)}, \mathbf{x}^{(v)}, p(\mathbf{z}^{(v)}) \log \frac{p(\mathbf{z}^{(v)}|\mathbf{x}^{(u)}, \mathbf{x}^{(v)})}{r^v(\mathbf{z}^{(v)}|\mathbf{x}^{(u)})} d\mathbf{x}^{(u)} d\mathbf{x}^{(v)} d\mathbf{z}^{(v)} \\
&= \mathbb{E}_{\mathbf{x}^{(u)}, \mathbf{x}^{(v)} \sim p(\mathbf{x}^{(u)}, \mathbf{x}^{(v)})}[D_{KL}(p(\mathbf{z}^{(v)}|\mathbf{x}^{(v)})||r^v(\mathbf{z}^{(v)}|\mathbf{x}^{(u)}))],
\end{aligned}
\tag{19}
$$

### A.3 PoE FUSION

Given the distribution $\mathcal{N}(\mu_v, \Sigma_v), v \in \mathcal{V}$ of each sub-expert for PoE fusion, the formulation of PoE fusion is as follows:

$$
\begin{aligned}
\mu_{poe} &= \frac{\sum_{v \in \mathcal{V}} \mu_v \frac{1}{\Sigma_v}}{\sum_{v \in \mathcal{V}} \frac{1}{\Sigma_v} + 1}, \\
\Sigma_{poe} &= \frac{1}{\sum_{v \in \mathcal{V}} \frac{1}{\Sigma_v} + 1},
\end{aligned}
\tag{20}
$$

where $\mu_{poe}$ and $\Sigma_{poe}$ are the fused mean and variance of multiple views, respectively. $\mu_v$ and $\Sigma_v$ mean the $v$-th view's mean and variance, respectively. Then, we have $p(\mathbf{z}|\{\mathbf{x}^{(v)}\}_{v \in \mathcal{V}}) \sim \mathcal{N}(\mu_{poe}, \Sigma_{poe})$.

### A.4 STATISTICS FOR FIVE DATASETS

In this section, we present details of the five databases used in our experiment in Table 2. The introductions of five widely used datasets are as follows:

1. **Corel5k** Duygulu et al. (2002): The Corel5k dataset contains 4,999 images and 260 annotations, with each image labeled by 1 to 5 tags.

2. **Pascal07** Everingham et al. (2009): PASCAL VOC 2007 is a widely used image dataset for visual object detection and recognition. In our experiments, we use 9,963 images spanning 20 object categories.

3. **ESPGame** Von Ahn & Dabbish (2004): The ESPGame dataset comprises 20,770 images collected from online interactive games, with 1 to 15 labels extracted per image. On average, it has 4.69 semantic labels per image and includes 268 unique labels in total.

4. **IAPRTC12** Henning et al. (2006): IAPRTC12 is a large-scale dataset with 19,627 images across 291 categories. Each image has up to 23 labels, extracted from the slogans or subtitles appearing in the image.

5. **Mirflickr** Huiskes & Lew (2008): The Mirflickr-25k open evaluation project consists of 25,000 images downloaded from Flickr, with 38 labels used in our experiments.

Table 2: Detailed information about five multi-view multi-label datasets in our experiments.

| Dataset | # Sample | # Label | # View | # Label/#Sample |
|---------|----------|---------|--------|-----------------|
| Corel5k | 4999 | 260 | 6 | 3.40 |
| IAPRTC12 | 19627 | 291 | 6 | 5.72 |
| ESPGame | 20770 | 268 | 6 | 4.69 |
| Pascal07 | 9963 | 20 | 6 | 1.47 |
| MIRFLICKR | 25000 | 38 | 6 | 4.72 |

## A.5 STATISTICS FOR EIGHT COMPETITORS

In this section, we give details of the eight comparison methods in Table 3.

Table 3: Simple information of nine comparison methods. 'Multi-view' denotes the method is designed for multi-view data; 'Missing-view' and 'Missing-label' represent their compatibility with missing views and missing labels.

| Method | Sources | Multi-view | Missing-view | Missing-label |
|--------|---------|------------|--------------|---------------|
| CDMM | KBS '20 | ✓ | ✗ | ✗ |
| DM2L | PR '21 | ✗ | ✗ | ✓ |
| LVSL | TMM '22 | ✓ | ✗ | ✗ |
| iMVWL | IJCAI '18 | ✓ | ✓ | ✓ |
| NAIM3L | TPAMI '22 | ✓ | ✓ | ✓ |
| DICNet | AAAI '23 | ✓ | ✓ | ✓ |
| DIMC | TNNLS '23 | ✓ | ✓ | ✓ |
| MSLPP | Neur Netw '24 | ✓ | ✓ | ✓ |
| SIP | ICML '24 | ✓ | ✓ | ✓ |

## A.6 EXTRA EXPERIMENTAL RESULTS ON TWO FULL DATASETS.

In this section, we show the results of nine methods on two datasets without any missing views and labels in Fig. 4.

## A.7 VISUALIZE THE CONSISTENCY OF EMBEDDING FEATURES

To demonstrate the constraints of our permutation strategy on multi-view consistency, we calculate the KL divergence between the private embeddings corresponding to any six views and draw heat maps in Fig. 5. We calculate the KL divergence among six views $\{r^v(\mathbf{z}^{(v)}|\mathbf{x}^{(v)})\}_{v \in \mathcal{V}}$ of two samples at three training epochs. It can be seen from the figure that as the training progresses continuously, the distribution consistency among views gradually increases reaches a balance. With the improvement of the prediction performance of the model, the encoding of each view in the latent space is evolving towards semantic consistency. However, due to the existence of the reconstruction regularization, the KL divergence does not decrease indefinitely.

## A.8 ABLATION STUDY

To assess the contribution of each component in PCVE, we conduct ablation experiments focusing on the two terms in our information-bottleneck objective: the view-specific reconstruction loss $\mathcal{L}_{re}$ and the permutation consistency loss $\mathcal{L}_{pc}$. We report results on two representative datasets (Corel5k and Pascal07) under the doubly-incomplete setting (50% missing views and 50% missing labels), and use the same training protocol and evaluation metrics as in the main experiments. Specifically,

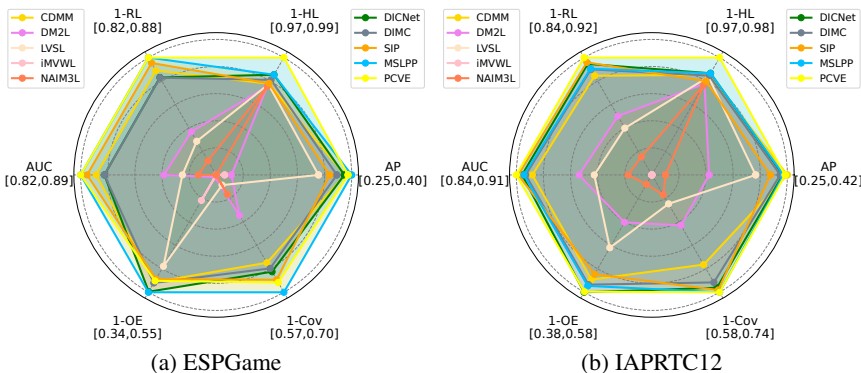

(a) ESPGame (b) IAPRTC12

Figure 4: Experimental results of nine methods on the two full datasets without any missing views or labels. The worst results are indicated at the center of radar chart, while the best results are represented by the vertexes, considering six evaluation metrics.

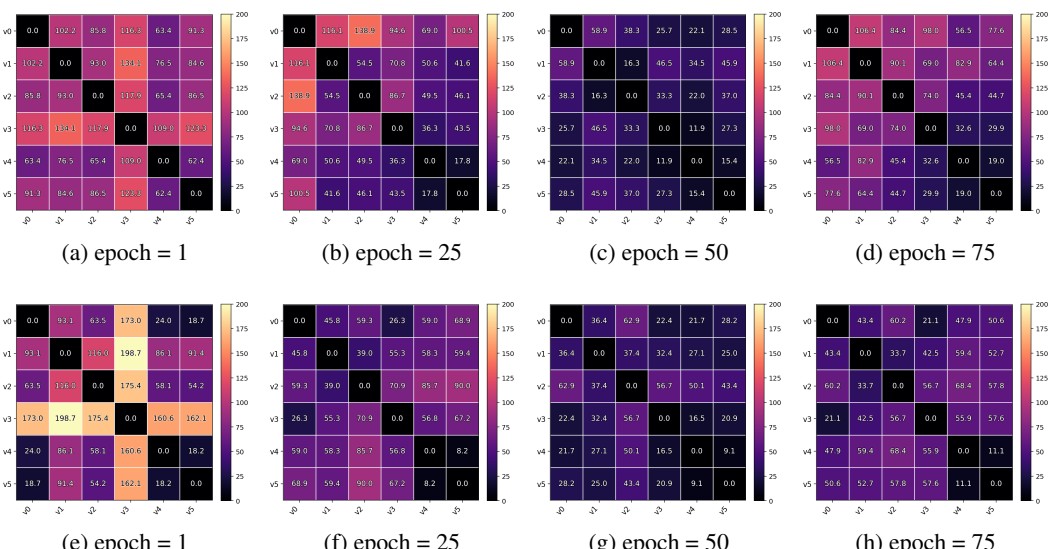

(a) epoch = 1 (b) epoch = 25 (c) epoch = 50 (d) epoch = 75

(e) epoch = 1 (f) epoch = 25 (g) epoch = 50 (h) epoch = 75

Figure 5: Visualization of KL divergence among views for two samples in three training epochs. (a)-(d) are for a sample and (e)-(h) are for another sample. "v1" denotes the view 1.

Table 4: Ablation results on Corel5k and Pascal07 datasets with 50% missing views and 50% missing labels. 'w/o' means 'without'. Loss $\mathcal{L}_{re}$ and $\mathcal{L}_{pc}$ denote the reconstruction term and permutation consistency constraint term, respectively.

| Method | Corel5k | | | | | | Pascal07 | | | | | |
|---|---|---|---|---|---|---|---|---|---|---|---|---|
| | AP | 1-HL | 1-RL | AUC | 1-OE | 1-Cov | AP | 1-HL | 1-RL | AUC | 1-OE | 1-Cov |
| PCVE w/o $\mathcal{L}_{re}$ & $\mathcal{L}_{pc}$ | 0.350 | 0.987 | 0.870 | 0.876 | 0.424 | 0.727 | 0.537 | 0.930 | 0.811 | 0.826 | 0.456 | 0.760 |
| PCVE w/o $\mathcal{L}_{pc}$ | 0.389 | 0.987 | 0.898 | 0.901 | 0.454 | 0.761 | 0.547 | 0.929 | 0.825 | 0.847 | 0.464 | 0.777 |
| PCVE w/o $\mathcal{L}_{re}$ | 0.261 | 0.987 | 0.853 | 0.858 | 0.289 | 0.677 | 0.511 | 0.931 | 0.788 | 0.815 | 0.441 | 0.740 |
| PCVE | 0.421 | 0.988 | 0.910 | 0.913 | 0.493 | 0.790 | 0.559 | 0.934 | 0.834 | 0.857 | 0.471 | 0.790 |

we remove $\mathcal{L}_{pc}$ and $\mathcal{L}_{re}$, respectively, and name them as "PCVE wo $\mathcal{L}_{pc}$" and "PCVE wo $\mathcal{L}_{re}$" in Table 4

From Table 4, it can be observed that removing $\mathcal{L}_{pc}$ consistently degrades the performance on all metrics. Without permutation consistent alignment, the model retains more view-private information and exhibits weaker cross-view generalization. Besides, dropping $\mathcal{L}_{re}$ leads to unstable training

and notable declines in AP and AUC. Over-aggressive compression causes information underflow: while cross-view posteriors appear compact (lower $\mathcal{L}_{pc}$), the shared representation becomes uninformative for accurate prediction. The full model yields the best overall performance, indicating that $\mathcal{L}_{re}$ supplies view-valid content to prevent collapse, while $\mathcal{L}_{pc}$ suppresses redundancy and enforces cross-view semantic invariance. The two terms are complementary in maintaining a compact yet sufficient representation.

## A.9 TIME COST STUDY

To assess the training and inference efficiency of PCVE, we report the training and testing time of ten methods on the Corel5k dataset (70% training split) in Table 5. Because model training time is highly sensitive to convergence criteria, we measure all methods under their default convergence settings. For single-view methods, we record the total training time summed over all views, and the inference time for a single view. For DICNet, SIP, and PCVE, we conduct 100 epochs for the training phase.

Table 5: Time cost of training and inference phases on the Corel5k dataset with 70% training samples. (Unit: s)

| Phase \ Method | CDMM | DM2L | LVSL | iMVWL | NAIML | DICNet | DIMC | MSLPP | SIP | PCVE |
|---|---|---|---|---|---|---|---|---|---|---|
| Training | 16.02 | 713.37 | 63.73 | 165.82 | 143.63 | 313.89 | 141.85 | 4889.84 | 336.11 | 412.21 |
| Inference | 1.73 | 0.04 | 0.64 | 0.02 | 0.01 | 0.05 | 0.04 | 0.05 | 0.01 | 0.03 |

## A.10 LIMITATIONS

Although PCVE shows strong effectiveness for iM3C and sheds light on aligning cross-view semantics via early permutation-consistent regularization, several limitations remain. First, our approach assumes that shared information suffices for prediction. This assumption may be strained under highly heterogeneous modalities (e.g., vision–language–audio) where view-private cues can be indispensable. Second, while permutation-based alignment reduces complexity, its stochastic matching scheme may introduce variance and could underperform with severely imbalanced or low-quality views. Third, our information bottleneck relies on variational bounds and KL-based surrogates; more accurate or adaptive mutual-information estimators might further improve stability and fidelity. Finally, we evaluate on established benchmarks with controlled missingness. Broader validation on real-world large-scale deployments with structured missing patterns and distribution shifts is needed to fully assess robustness and generality.

