# OpenReview forum: "Permutation-Consistent Variational Encoding for Incomplete Multi-View Multi-Label Classification"
_ICLR.cc/2026/Conference — ICLR 2026 Poster_

### Official Review · Reviewer_jgRE · 2025-10-26

**Soundness:** 4
**Presentation:** 3
**Contribution:** 4
**Rating:** 8
**Confidence:** 5

**Summary:**

This paper introduces Permutation-Consistent Variational Encoding (PCVE), a novel framework for addressing the challenges of incomplete multi-view multi-label classification (iM3C). By leveraging information bottleneck (IB) principles with a permutation-consistency regularization, the framework promotes semantic alignment across views while mitigating redundancy and over-alignment. The framework is extensively evaluated on five multi-label benchmarks under both incomplete and complete settings, achieving SOTA performance.

**Strengths:**

1.	The paper introduces a novel permutation-consistency objective that aligns cross-view latent representations while reducing computational complexity, offering a scalable and effective solution for semantic alignment.
2.	The proposed framework balances view-specific information preservation and cross-view semantic consistency well, avoiding over-compression and ensuring task-relevant representation adequacy.
3.	PCVE achieves SOTA results across multiple metrics in both doubly incomplete and fully observed settings, demonstrating its robustness and generality.

**Weaknesses:**

1.    The permutation-consistency mechanism and information bottleneck formulation are innovative, but the novelty over existing methods (e.g., DICNet, NAIM3L) is not sufficiently clarified. Additionally, the framework’s reliance on PoE-based fusion and variational inference may be similar to prior multi-view learning approaches, like SIP.
2.	Some statements are unclear, such as "we randomly select view indices from the available view set V without replacement." What does this mean?
3.	Parameter sensitivity experiments on alpha seem to be missing, do the authors have any reason to clarify it further?
4.	The dataset descriptions lack sufficient detail, making it difficult to assess their relevance to iM3C tasks. Authors are advised to provide more details about the datasets.

**Questions:**

See the Weaknesses.

---

> ### Author Response · Authors · 2025-11-20
> **Response to Reviewer jgRE**
>
> Thank you for your recognition of our work. We are deeply impressed by your professional review comments.
>
> **W1**: Thank you very much for your comment, we further clarify our innovativeness: (1) Unlike prior methods that align all views via global and all-pairs distances or shared-coding coupling, PC aligns via randomized one-to-one permutations with a soft KL regularizer. This avoids over-alignment seen in dense all-pairs KL and yields controlled similarity constraint when combined with reconstruction and view-specific feature preservation. (2) Our objectives are derived from an IB perspective that jointly regulates: information preservation $I(z; y)$; cross-view compression $I(x^{(v)}; z^{(v)})$; and permutation-consistent regularization as an explicit constraint on posterior families.
>
> **W2**: We aim to ensure diversity in cross-view encoding sources. For each sample with available-view set $\mathcal{V}$, we form a permutation $\pi$ over $\mathcal{V}$ so that each destination view is paired with a distinct source view. “Without replacement” means we do not pick the same view index more than once when constructing $\pi$. Every selected index is unique within that permutation. For example, supposing the available views are $\mathcal{V}$ = {1, 2, 4, 6}, a valid permutation could be $\pi$ = {4, 1, 6, 2}. Here, each source index (4,1,6,2) is used exactly once. In contrast, selections like {4,4,6,6} would be invalid because they reuse indices (with replacement), reducing source diversity.
>
> **W3**: Thank you for the suggestion. In our IB-derived objective, the α-weighted term for maximizing $I(z; y)$ is instantiated via the supervised classification loss, while the compression side is decomposed into $L_{pc}$ (permutation-consistency) and $L_{re}$ (reconstruction). In practice, we control the balance between alignment and reconstruction with a single coefficient $\beta$ after this decomposition. Hence, our sensitivity analysis focuses on $\beta$, which directly governs the trade-off most relevant to the bottleneck behavior.
>
> **W4**: The dataset descriptions lack sufficient detail, making it difficult to assess their relevance to iM3C tasks. Authors are advised to provide more details about the datasets.
>
>
> *(1) Corel5k: The Corel5k dataset contains 4,999 images and 260 annotations, with each image labeled by 1 to 5 tags.
> (2) Pascal07\cite{everingham2009pascal}: PASCAL VOC 2007 is a widely used image dataset for visual object detection and recognition. In our experiments, we use 9,963 images spanning 20 object categories.
> (3) ESPGame: The ESPGame dataset comprises 20,770 images collected from online interactive games, with 1 to 15 labels extracted per image. On average, it has 4.69 semantic labels per image and includes 268 unique labels in total.
> (4) IAPRTC12: IAPRTC12 is a large-scale dataset with 19,627 images across 291 categories. Each image has up to 23 labels, extracted from the slogans or subtitles appearing in the image.
> (5) Mirflickr: The Mirflickr-25k open evaluation project consists of 25,000 images downloaded from Flickr, with 38 labels used in our experiments.*

---

### Official Review · Reviewer_wLuJ · 2025-10-31

**Soundness:** 3
**Presentation:** 3
**Contribution:** 3
**Rating:** 6
**Confidence:** 5

**Summary:**

The paper proposes a PCVE framework to address the dual incompleteness problem in multi-view multi-label learning. The idea is to use the permutation consistency regularization to constrain the distribution of latent variables of different views in the encoding stage, so that the shared semantics across views are aligned. At the same time, the reconstruction term is used to retain effective information, and PoE is used to fuse to form a shared representation for multi-label prediction. The authors give the target decomposition from the IB perspective and the corresponding variational upper and lower bounds, claiming that it can outperform a variety of strong baselines at different missing rates. The highlight of the work is to pre-place the consistency constraint in the encoding stage and introduce the distribution cluster exchange of random permutation to reduce complexity.

**Strengths:**

1. The problem is highly motivated for the practically important dual incomplete scenario. The complexity of this problem has not been considered in existing research, and this paper grasps this difficulty and helps real-world applications.

2. The methods proposed in this paper have a unified form, such as encoding consistency, view-private information reconstruction, PoE fusion and masked multi-label learning, and the overall design is self-consistent.

3. This article systematically derives the objectives from IB and provides the variational bounds and training objectives that can be implemented for optimization.

**Weaknesses:**

1. Will the random permutation proposed by the authors introduce additional noise? Also, will swapping them at each training epoch affect the stability of the training?

2. The $z^1$ in Figure 1 does not match the description in the text, which may cause ambiguity. I suggest that the authors revise Figure 1 to keep the symbols consistent.

3. There is a lack of sufficient explanation on how missing views are handled. How did the authors handle missing views during the encoding stage? I believe that the way to handle missing view is an important step in the method of this paper.

4. What is the scope of application of PCVE proposed in this paper? In real applications, can it handle heterogeneous multimodal data?

5. Although the computational complexity has been greatly reduced thanks to the proposed random permutation strategy, I am still interested in the overall parameter count and computational efficiency because it is different from the conventional VAE framework based on single modality encoding.

**Questions:**

Please refer to the weaknesses for my considerations and concerns.

---

> ### Author Response · Authors · 2025-11-20
> **Response to Reviewer wLuJ**
>
> Thank you for your valuable comments. We respond to this as follows:
>
> **W1**: We pre-generate multiple permutation schedules and cycle them across epochs (without replacement within a batch). This yields short-term gradient variability but stable long-term descent; loss curves remain smooth and convergent in our experiments. Empirically, across five datasets we observed no divergence; the permutation acts as beneficial regularization rather than destabilizing noise.
>
> **W2**: Thank you for pointing this out. We have revised Figure 1 to replace $z^1$ with $z^{(1)}$ to match the notation in the text, and we have checked all other symbols in the figure to ensure consistency throughout the manuscript.
>
> **W3**: Thank you for the suggestion. First, we denote the observed views by $\mathcal{V}$. All encoders, permutation pairing, and fusion operate only on $ v \in \mathcal{V}$. In our framework. we adopt free-imputation method instead of imputing missing views. For example, we form permutations over the available views only and compute $r^v(z^{(v)}|x^{(v)})$ for $v ∈ \mathcal{V}$; no terms are created for missing views. In the reconstruction $L_{re}$ is computed only for available views.
>
> **W4**: Thanks for your question. PCVE targets incomplete multi-view multi-label settings where some views and/or labels are missing. It applies to (1) fully observed, partially observed, and severely incomplete views; (2) Missing labels (3) Different numbers of views across samples; (4) Both training and inference with arbitrary available-view subsets. Our permutation-consistency is a soft KL regularizer balanced by supervision and reconstruction, encouraging semantic alignment without collapsing modality-specific signals. The framework supports uncertainty-aware weights in both PoE and the consistency term to account for differing modality qualities. In summary, PCVE is broadly applicable to incomplete multi-view learning and is directly extendable to heterogeneous multimodal applications with appropriate encoders and lightweight reconstruction objectives.
>
> **W5**: Thanks for your comment! Each view-to-view stochastic encoder head is a shallow MLP (Gaussian mean/variance heads). Despite having m×m heads, their depths are small, so the total parameter overhead is modest (about 30M). We have supplemented the training and reasoning time for each method in the appendix:
>
> *To assess the training and inference efficiency of PCVE, we report the training and testing time of ten methods on the Corel5k dataset (70\% training split) in Table 5. Because model training time is highly sensitive to convergence criteria, we measure all methods under their default convergence settings. For single-view methods, we record the total training time summed over all views, and the inference time for a single view. For DICNet, SIP, and PCVE, we conduct 100 epochs for the training phase.*
>
> | Phase | CDMM | DM2L | LVSL | iMVWL | NAIML | DICNet | DIMC | MSLPP | SIP | PCVE |
> |-----------|-------|--------|------|-------|-------|--------|-------|--------|-------|-------|
> | Training | 16.02 | 713.37 | 63.73| 165.82| 143.63| 313.89 | 141.85| 4889.84| 336.11| 412.21|
> | Inference | 1.73 | 0.04 | 0.64 | 0.02 | 0.01 | 0.05 | 0.04 | 0.05 | 0.01 | 0.03 |

---

### Official Review · Reviewer_gaes · 2025-10-31

**Soundness:** 3
**Presentation:** 3
**Contribution:** 3
**Rating:** 6
**Confidence:** 5

**Summary:**

This paper tackles the double-missing problem in incomplete multi-view and multi-label (iM3C) learning by proposing a Permutation-Consistent Variational Encoding (PCVE) framework based on the Information Bottleneck principle. To suppress view-private redundancy and prevent over-alignment, the framework introduces a scalable permutation consistency regularization mechanism via "distributed cluster swapping" with random permutations. Additionally, reconstruction terms are incorporated to preserve essential information and avoid representation collapse. Extensive experiments under various benchmarks and missing ratios verify that the proposed approach achieves consistent improvements over existing state-of-the-art techniques.
.

**Strengths:**

1. This work innovatively advances cross-view consistency to the encoding stage and introduces a "permutation consistency" regularization strategy, which effectively achieves efficient alignment while balancing sufficiency and compactness of representations. The proposed method is intuitive with a unified framework.
2. The objective decomposition and variational lower/upper bounds are derived from the Information Bottleneck perspective, which is clearly motivated and well defined. (e.g., the motivation for the minimum sufficiency of conditional mutual information).
3. The authors conduct extensive evaluations across five standard datasets and six metrics, considering both complete and double-missing scenarios. Comparison against nine strong baselines demonstrates stable and leading performance.
4. The readability of this paper is good, with clear motivations and strong experimental results supporting.

**Weaknesses:**

1. Suppose the mutual information of each view about y is equal, and "the shared information is sufficient for prediction". It may not be suitable in strong heterogeneous modes or tasks (such as visual-language-audio).
2. In this paper, each view needs to construct the encoder sub-distribution $r_v^n$ to each target view, but there is a lack of analysis regarding the number of model parameters, training time, etc. I suggest the authors supplement the relevant experiments.
3. The explanation of some symbols needs to be strengthened, such as what is the relationship between $p$ and $p_i$. The lack of a clear explanation will lead to confusion among readers.

**Questions:**

1. In PoE fusion and consistency regularization, are unequal weights of views considered? For example, the weight of low-quality views with large variance is reduced.
2. How does the PCVE handle missing labels? It seems that this part (handle missing labels) should be clarified.
3. There is a typo in line 59, "IIn".

---

> ### Author Response · Authors · 2025-11-20
> **Response to Review gaes**
>
> We are very grateful for your feedback and we reply as follows:
>
> **W1**: Thanks for the comment! We view the assumption as semantic-level consistency, not low-level similarity. Across modalities, the same instance expresses common label-relevant semantics. PCVE explicitly targets this shared semantic subspace while allowing view-specific feature via reconstruction. Permutation-consistency is a KL regularizer, balanced by reconstruction ($L_{re}$) and supervision ($L_{ce}$). It encourages convergence only along task-relevant dimensions and modality-private cues are preserved if they aid prediction. The framework’s goal, approximating semantic consistency across views, extends naturally to multimodal learning. It leverages the existence of a non-empty shared semantic core and uses supervised loss and reconstruction to retain modality-specific predictive information and gain the cross-view semantic consistency.
>
> **W2**: Thanks for the comment! Each view-to-view stochastic encoder head is a shallow MLP (Gaussian mean/variance heads). Despite having m×m heads, their depths are small, so the total parameter overhead is modest (about 30M). We have supplemented the training and reasoning time for each method in the appendix:
>
> *To assess the training and inference efficiency of PCVE, we report the training and testing time of ten methods on the Corel5k dataset (70\% training split) in Table 5. Because model training time is highly sensitive to convergence criteria, we measure all methods under their default convergence settings. For single-view methods, we record the total training time summed over all views, and the inference time for a single view. For DICNet, SIP, and PCVE, we conduct 100 epochs for the training phase.*
>
> | Phase | CDMM | DM2L | LVSL | iMVWL | NAIML | DICNet | DIMC | MSLPP | SIP | PCVE |
> |-----------|-------|--------|------|-------|-------|--------|-------|--------|-------|-------|
> | Training | 16.02 | 713.37 | 63.73| 165.82| 143.63| 313.89 | 141.85| 4889.84| 336.11| 412.21|
> | Inference | 1.73 | 0.04 | 0.64 | 0.02 | 0.01 | 0.05 | 0.04 | 0.05 | 0.01 | 0.03 |
>
> **W3**: We apologize for the confusion and have clarified the notation in the revised manuscript. Specifically: $p$ denotes the vector of predicted probabilities over all $C$ labels, i.e., p ∈ [0, 1]^C. p_i denotes the i-th component of p, i.e., the predicted probability for label i (1 ≤ i ≤ C).
>
> **Q1**: As shown in Eq. (18), the fused covariance is $\Sigma_{poe}=\frac{1}{\sum_{v\in \mathcal{V}}\frac{1}{\Sigma_v}+1}$ and the fused mean uses precision weights ($\Sigma_v^{-1}$). Hence, a view with larger variance (higher uncertainty/lower quality) receives a smaller weight in the fused mean. Practically, noisy views contribute less to the joint posterior.
>
> **Q2**: We maintain an available label set $\mathcal{G}$ per sample. In Eq. (10), the multi-label cross-entropy is computed only over $i \in \mathcal{G}$. This mask prevents the model from being penalized for unobserved labels, reducing bias from incomplete supervision while still leveraging all available annotations.
>
> **Q3**: Thank you for catching this. We have carefully proofread the manuscript and corrected this typo (“IIn” → “In”) along with other minor typographical issues.

---

### Official Review · Reviewer_RiXu · 2025-10-31

**Soundness:** 2
**Presentation:** 3
**Contribution:** 2
**Rating:** 4
**Confidence:** 4

**Summary:**

This paper explores the challenging problem of multi-view and multi-label classification with both view and label incompleteness. The authors propose a framework named Permutation-Consistent Variational Encoding (PCVE), which incorporates an information bottleneck strategy to learn variational representations that can aggregate shared semantics across views. PCVE maximizes a variational evidence lower bound (ELBO) to preserve task-relevant information and introduces a permutation-consistency regularization to encourage distributional alignment between encoded representations of the same semantic content across different views.

**Strengths:**

- The work defines an interesting and meaningful learning setting, incomplete multi-view missing multi-label classification (iM3C).
- The proposed universal variational encoding framework is flexible and theoretically capable of handling arbitrary missing-view and missing-label patterns.
- The idea of permutation consistency provides a potentially simple yet effective strategy for encouraging cross-view semantic consistency.

**Weaknesses:**

1. The experimental results are not convincing and the novelty of this paper is not supported by the experiments. In Table 1, PCVE achieve trivial gains against other methods.
2. The paper is difficult to follow. For example, Fig. 1 is neither properly referenced nor introduced in the main text, which harms overall readability.
3. The derivations are extensive and dense; many could be moved to supplementary materials. Some portions appear overly “formulaic,” potentially obscuring the main conceptual contributions.
4 Though the paper emphasizes learning shared semantic information, there is no intuitive or visual analysis of the latent variable z(v).
5. Stochasticity Risk in Permutation Strategy. While random permutation reduces computational complexity, it may introduce training instability. This could be particularly problematic with severely imbalanced view quality (e.g., pairing a high-quality view with a low-quality/noisy one, potentially hindering convergence or leading to sub-optimal performance.
6 Limitations in Experimental Setup.  The missing data pattern is randomly generated, failing to simulate the more common real-world scenario of non-random missingness (e.g., where the absence of a view correlates with sample characteristics), which limits the demonstration of robustness in practical applications.

**Questions:**

1 Regarding Assumptions and Information Loss: Assumption 2.1 posits that all views havethe same mutual information with the target, When this assumption doesn't hold in practice(e.g. one view is significantly more discriminative), how does PCVE prevent valuable, unique semantic information from this "strong view" from being prematurely suppressed ordiscarded during the permutation-consistency process?
2 Comparison of Alignment Mechanisms: Compared to traditional alignment methods thatdirectly minimize distribution distance (e.g., KL divergence) in the latent space, could youprovide more direct experimental evidence (e.g., visualizations or quantitative metrics) todemonstrate the specific advantages of permutation consistency in avoiding "over-alignment" and preserving necessary diversity?
3 Real-World Missing Patterns: The current experiments use random missingness. Howwould PCVE's performance be affected if the missing pattern were non-random (e.g. theabsence of a certain view is strongly correlated with samples of specific categories)? What isyour view on the robustness of your method to such bias?
4 Applicability beyond image features: Has the approach been validated on truly multimodaldata (e.g., image-text or audio-visual tasks)? lf not, what challenges would arise whenextending PCVE to such heterogeneous domains?

---

> ### Author Response · Authors · 2025-11-20
> **Response to Review RiXu**
>
> **W1**: Thanks for your feedback! We compare against the latest, state-of-the-art iM3C methods (DICNet, DIMC, MSLPP, SIP, etc.). These methods have already taken a clear lead over the earlier work in their respective papers. Therefore, achieving a "comprehensive victory" on such a strong baseline itself not only poses statistical and methodological difficulties but also better demonstrates the effectiveness of the method. Besides, we avoid heavy, task-specific tricks (e.g., complex pseudo-labeling). Gains come from the proposed permutation-consistent IB framework itself, highlighting methodological contribution rather than engineering. Under fully observed data (Fig. 2), PCVE still leads, indicating benefits are not tied to a specific incompleteness setup, showing better generality. The ablation results (Table 4) directly validate the core idea, i.e., removing the permutation-consistency term $L_{pc}$ or the reconstruction term $L_{re}$ degrades performance, and removing both leads to the largest drop. This shows the gains stem from the proposed permutation-consistent alignment within the IB framework, not incidental training artifacts.
>
> **W2**: 2.	Thank you very much for your meticulous review. We agree that Fig. 1 was insufficiently referenced in main text. We have revised the main text to explicitly introduce it at the start of Section 3:
>
> *In this section, we detail the proposed permutation-consistent variational encoding (PCVE) framework. The overall pipeline is illustrated in Fig. 1: the top panel depicts multi-view shared information learning and reconstruction, while the bottom panel shows cross-view fusion and multi-label classification.*
>
> **W3**: Thanks for the comment. We keep several intermediate steps to avoid discontinuities in the logic. For example, in Eq. (5) we briefly expand $I(x^{(v)}; z^{(v)})$ and present an intermediate form before the final lower bound specifically to introduce the variational encoder $q^v$. If the intermediate steps are removed, the lower bound obtained by Eq. (5) will be confusing. The same applies to Eqs. (6) and (8), where the intermediate manipulations clarify how the conditional mutual information is upper-bounded via KL terms. We only retain the minimal intermediate steps needed to make the transitions transparent and the full line-by-line derivations are provided in the appendix. Readers seeking the core ideas can follow the main text, while those needing full mathematical details can refer to the supplement.
>
> **W4 and Q2**: Thank you for the suggestion. We have added an intuitive visualization to the appendix to evidence cross-view semantic consistency in the latent space. We track pairwise KL among view posteriors over different training epochs. As cross-view consistency increases, KL divergences tend to be stable rather than collapse to zero. $L_{re}$ counteracts further contraction, stabilizing a non-zero similarity level. This indicates a balance: shared semantics are aligned, while view-specific content is preserved. The following analysis is added in the appendix:
>
> *To demonstrate the constraints of our permutation strategy on multi-view consistency, we calculate the KL divergence between the embeddings corresponding to any six views and draw heat maps in Fig. 5. We calculate the KL divergence among six views $\{r^v(\mathbf{z}^{(v)}|\mathbf{x}^{(v)})\}_{v\in \mathcal{V}}$ of two samples at four training epochs. It can be seen from the figure that as the training progresses continuously, the distribution consistency among views gradually increases and reaches a balance. With the improvement of the prediction performance of the model, the encoding of each view in the latent space is evolving towards semantic consistency. However, due to the existence of the reconstruction regularization, the KL divergence does not decrease indefinitely.*
>
> **W5**: Thank you for the comments. We pre-generate multiple permutation schedules and cycle them across training. This introduces short-term gradient direction variability but yields a stable long-term descent toward cross-view posterior alignment. The multi-label objective provides an external semantic anchor. When a high-quality view is matched with a low-quality/noisy view, the shared classifier loss steers both encoders toward task-relevant structure; in practice, the higher-quality view serves as a teacher signal that lifts the weaker one rather than being dragged down.

---

> ### Author Response · Authors · 2025-11-20
> **The following response**
>
> **W6 and Q3**: Thank you for your professional comment. In our paper, we follow the related evaluation setup in iM3C field, which uses random missingness at controlled rates to enable fair comparison with prior work and strong baselines. It should be pointed out that most existing multi-view multi-label datasets are fully observed; naturally incomplete multi-view multi-label datasets are scarce, especially with label incompleteness. Random masking remains a pragmatic proxy to stress-test methods across a range of missing rates. In future work, we will be committed to collecting or constructing benchmarks with structured/non-random missingness to simulate the real-world scenario.
>
> **Q1**: Thanks for your question. In our learning objective, permutation-consistency KL ($L_{pc}$: compresses view-private noise by nudging view posteriors closer; supervised multi-label classification loss ($L_{ce}$) + view reconstruction ($L_{re}$): anchors learning to task semantics and preserves view-valid content. These forces are complementary, i.e., if aligning view A with better discrimination toward a noisier view B would harm prediction, $L_{ce}$ and $L_{re}$ will push it back, preventing semantic information in A from being erased. In practice, weaker views produce higher-variance posteriors; under PoE fusion, higher-uncertainty experts contribute less to the joint posterior. Thus, the strong views naturally dominate the fused representation, mitigating the risk of being “dragged down.”
>
> **Q4**: We acknowledge that publicly available multi-view multi-label datasets with heterogeneous modalities (e.g., image–text, audio–video) are scarce, and we have not yet included such benchmarks. We are actively constructing multimodal multi-label datasets and plan to report these in follow-up work. PCVE is modality-agnostic at the framework level. Each view only needs a probabilistic encoder $r^v(z^{(v)}|x^{(v)})$ and a decoder $q^v(x^{(v)}|z^{(v)})$. Thus, image, text, audio, and video can be accommodated by plugging in appropriate front-end encoders to produce latent distributions.
>
> Thank you again for your valuable comments and suggestions. Your comments are crucial for us to improve the quality of our papers. We hope our reply can address your concerns. We would be happy to answer any further questions you may have.

---

### Author Response · Authors · 2025-11-29
**Rebuttal summary**

**We thank the reviewers for their thoughtful feedback.** Your efforts have contributed to the improvement of article quality, and your professional attitude and constructive comments have advanced the ICLR community. Here we summarize the reviewers' comments and our responses.

All the reviews acknowledge that our paper tackles a practically important iM3C setting with a unified variational framework that is theoretically sound and flexible for arbitrary missing views and labels. The idea of permutation-consistency mechanism is innovative and
extensive experiments verified the effectiveness of our PCVE.  They also agree that the readability and clarity are both excellent.

The common concerns are focused on (1) Visualizations regarding the effectiveness of permutation-consistency and IB framework on eliminating over-alignment risk; (2) A clearer explanation regarding handling missingness and unequal view quality; (3) Computational Efficiency and Cost. In response to these concerns, we have added an intuitive visualization to the appendix to evidence cross-view semantic consistency, clarified imputation-free handling of missing views and masked loss for missing labels, and reported a modest parameter count and runtime efficiency analysis.

**We believe that with the efforts of ourselves and the reviewers, the quality of our article has been further improved. We thank the reviewers and the AC again for their hard work.**

---

### Meta-Review · Area_Chair_wrqV · 2026-01-06

**Summary:**

This paper presents Permutation-Consistent Variational Encoding (PCVE), a unified variational framework for incomplete multi-view multi-label classification (iM3C) that integrates an information bottleneck strategy, permutation-consistency regularization, PoE fusion, and masked multi-label learning to handle arbitrary missing views and labels. Reviewers recognized its practical significance in addressing the iM3C setting, innovative permutation-consistency mechanism for cross-view semantic alignment, solid theoretical foundation derived from the Information Bottleneck principle. The authors actively responded to and addressed the main concerns, including supplementing and clarifying the imputation-free handling of missing views and masked loss for missing labels, and elaborating on permutation strategy stability. Based on the above considerations, I recommend acceptance.

**Reviewer Concerns:**

All the main concerns were addressed by the rebuttal.

**Reviewer Scores:**

Reviewer RiXu would have raised the score from 4 to 6. For the other reviewers, I believe their scores would have remained unchanged

---

### Decision · Program_Chairs · 2026-01-26

Accept (Poster)